# High protein copy number is required to suppress stochasticity in the cyanobacterial circadian clock

Justin Chew [1], Eugene Leypunskiy [2], Jenny Lin[3], Arvind Murugan [4] & Michael J. Rust[4,5]

Circadian clocks generate reliable ~24-h rhythms despite being based on stochastic biochemical reactions. The circadian clock in *Synechococcus elongatus* uses a post-translational oscillator that cycles deterministically in a test tube. Because the volume of a single bacterial cell is much smaller than a macroscopic reaction, we asked how clocks in single cells function reliably. Here, we show that *S. elongatus* cells must express many thousands of copies of Kai proteins to effectively suppress timing errors. Stochastic modeling shows that this requirement stems from noise amplification in the post-translational feedback loop that sustains oscillations. The much smaller cyanobacterium *Prochlorococcus* expresses only hundreds of Kai protein copies and has a simpler, hourglass-like Kai system. We show that this timer strategy can outperform a free-running clock if internal noise is significant. This conclusion has implications for clock evolution and synthetic oscillator design, and it suggests hourglass-like behavior may be widespread in microbes.

[1] Medical Scientist Training Program, Pritzker School of Medicine, University of Chicago, 900 E 57th St, Chicago, IL 60637, USA. [2] Graduate Program in Biophysical Sciences, University of Chicago, 900 E 57th St, Chicago, IL 60637, USA. [3] Department of Biochemistry and Molecular Biology, University of Chicago, 900 E 57th St, Chicago, IL 60637, USA. [4] Department of Physics, University of Chicago, 900 E 57th St, Chicago, IL 60637, USA. [5] Department of Molecular Genetics and Cell Biology, University of Chicago, 900 E 57th St, Chicago, IL 60637, USA. Correspondence and requests for materials should be addressed to M.J.R. (email: mrust@uchicago.edu)

Circadian clocks are biochemical oscillators that enable organisms to anticipate the day-night cycle. Their utility depends on the ability to make accurate predictions about the future[1,2] and thus requires precise, deterministic timing. This precision must be achieved despite the fact that biochemical processes are composed of elementary reaction events, each of which occurs with stochastic timing. Indeed, most synthetic cellular oscillators produce noticeably irregular rhythms[3–5]. In contrast, natural circadian clocks can be extremely precise[6–8]. It is generally not known how biological clocks create deterministic rhythms from their stochastic components, or how the architecture of clock networks responds to the constraints of molecular noise.

To address these questions, we turned to the cyanobacterial circadian clock. Cyanobacteria are a diverse clade of photosynthetic prokaryotes that carry *kai* clock genes that generate daily oscillations in physiology[9–11]. The core mechanism of oscillation in the cyanobacterial clock is post-translational and can be reconstituted using purified proteins[12]. KaiA and KaiB modulate the autocatalytic activity of KaiC, producing self-sustaining rhythms of KaiB-KaiC binding and multisite phosphorylation on KaiC[13].

We present an experimental study of the coherence of circadian rhythms in single cells as the number of Kai protein molecules per cell is varied using an inducible expression system. We use a stochastic modeling approach to study the post-translational Kai reaction network, and we identify the delayed negative feedback loop that sequesters and inhibits KaiA as a bottleneck that amplifies molecular noise in the clock. Finally, we consider a simplified Kai system in the tiny cyanobacterium *Prochlorococcus* where the KaiA-dependent feedback loop is absent. Our analysis supports the hypothesis that internal noise tends to disfavor free-running behavior in the Kai system, suggesting that circadian clocks are disadvantageous under some conditions.

## Results

**The Kai clock must be highly expressed to function reliably.** Because the volume of a bacterial cell is smaller than the volume of a test-tube reaction by many orders of magnitude, we suspected that stochasticity due to finite numbers of clock proteins might be an important constraint in cells. To study this effect, we engineered a strain of the model cyanobacterium *Synechococcus elongatus* PCC 7942 where the copy numbers of the Kai proteins are under experimental control. We replaced the native copies of the *kai* genes with copies containing a theophylline-inducible riboswitch previously shown to modulate translational efficiency[14,15], allowing us to tune Kai protein expression (Fig. 1a, b). In vitro, the ratio of KaiA to KaiC must be kept within a specific range for oscillations to occur[16,17]. Thus, in our engineered strain, *kaiB* and *kaiC* are transcribed from a constitutive promoter and *kaiA* from an isopropyl β-D-1-thiogalactopyranoside (IPTG)-inducible promoter to allow independent control of KaiA expression (Fig. 1a). This system removes the natural transcriptional feedback in the system and allows us to focus on the core post-translational oscillator.

Using quantitative western blotting, we found that wild-type cells express ~4000 KaiA, ~11,000 KaiB, and ~8000 KaiC copies per cell. Our estimates for KaiB and KaiC are similar to a previous report[18], though our estimate for KaiA is markedly higher. The stoichiometry we observe here is similar to that needed to support oscillations with purified proteins[13]. We then determined that our engineered strain is capable of expressing Kai proteins in a range spanning from 100s up to 10,000s of copies per cell (Fig. 1c and Supplementary Fig. 1). To characterize the ability of this inducible system to produce circadian rhythms, we used a luciferase assay

to report population-level gene expression rhythms. We found that while high levels of theophylline induction produced wild-type-like rhythms, oscillations at the population level weakened or vanished at lower levels of induction even though Kai proteins were still expressed (Fig. 1d and Supplementary Fig. 2).

We reasoned that loss of population-level oscillations at lower Kai protein expression levels could be explained by two possibilities—rhythms could either be lost in individual cells, or they could persist in single cells but with significant desynchronization between cells. To distinguish between these scenarios, we used time-lapse fluorescence microscopy to observe single-cell rhythms in constant conditions (Supplementary Fig. 3A). Consistent with previous reports[19,20], we observed that circadian rhythms in single wild-type cells are remarkably precise with <5% timing error per clock cycle (standard deviation/mean of peak-to-peak times). When we analyzed our tunable expression strain, we found that single cells in fact maintained high-amplitude rhythms even at low levels of theophylline (Supplementary Fig. 3B, C), but these rhythms desynchronized over time between cells in a theophylline-dependent manner (Fig. 2a, b). 370 μM theophylline (~12,000 copies KaiC/cell) produced coherent single-cell rhythms comparable to wild type that maintained synchrony over one week, while 92 μM theophylline (~7000 copies KaiC/cell) led to rhythms that were markedly noisier, and 23 μM theophylline (~2600 copies KaiC/cell) produced very noisy rhythms where cells in the same microcolony appeared to adopt nearly random phases after a few days (Fig. 2, Supplementary Movies 1–4). We quantified the distribution of peak-to-peak times in these single cell movies (Fig. 2c), and found that, in addition to increased variability, the mean period of the oscillation increases at low expression levels. Both of these effects would tend to make the clock state mismatch the 24 h day-night cycle. Notably, even when grown in entraining light-dark cycles, cells at low theophylline conditions show evident stochasticity in oscillator phase (Supplementary Fig. 3D–E).

When protein expression level is reduced in these experiments, both protein copy number and concentration are reduced. Because the post-translational oscillator is highly robust to total protein concentration, we expect that copy number changes are the main driver of stochasticity[13,21] (Supplementary Fig. 4). To experimentally disentangle these effects, we used natural variability in cell size to stratify our analysis and focus on cells with unusually small or large volumes. Since protein concentration is relatively constant across cell sizes[22,23] (Supplementary Fig. 3F), we used cell volume as a proxy for copy number within each induction condition. We quantified relative peak-to-peak timing errors in these cells at different induction levels and found that shorter cells had noisier rhythms compared to longer cells at lower induction conditions (Fig. 2d). From these results, we conclude that high copy numbers of the Kai proteins are required to effectively suppress stochasticity in the circadian rhythm.

**Delayed negative feedback on KaiA amplifies internal noise.** How does the presence of many copies of the Kai proteins suppress timing errors, and what features of the oscillator circuit are most vulnerable to noise at low copy number? We first consider a simple result from the theory of stochastic chemical reactions: if a population of $N$ identical molecules transit independently through a sequence of $m$ identical reaction steps, we expect the time at which half of the molecules have completed the reaction to show timing errors proportional to $1/(m\sqrt{N})$ where the factor of $\sqrt{N}$ comes from averaging out uncorrelated fluctuations in timing between different molecules. In this case, protein copy numbers in the 1000s would be sufficient to achieve mean timing errors less than 5%. However, the above scenario does not

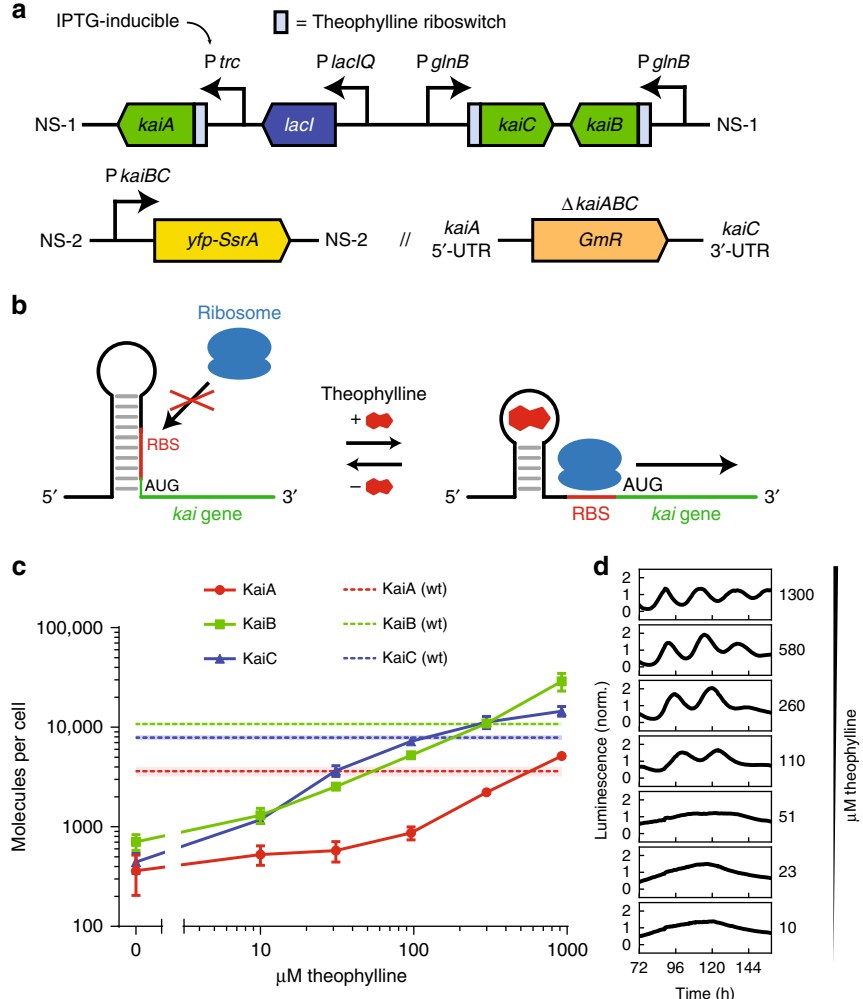

**Fig. 1** Characterization of the Kai copy-number tunable strain. **a** A theophylline riboswitch regulates translational efficiency of all three *kai* genes, and transcriptional regulation of *kaiA* is controlled by an IPTG-inducible promoter. Clock state is reported by EYFP-SsrA expressed from the *kaiBC* promoter. **b** Theophylline regulates translation by freeing the ribosome binding site upstream of each *kai* gene. **c** Kai copy numbers plotted as a function of theophylline concentration with 1 µM IPTG (solid line), and Kai copy numbers in wild-type cells (dotted line). Vertical error bars or shaded area indicate standard error of the mean from three replicates. **d** Colony-level oscillations detected with a bioluminescent reporter in the copy number tunable strain with 1 µM IPTG and various theophylline concentrations

describe a self-sustaining circadian oscillator; to achieve stable oscillations, feedback loops must couple the reactions of the molecules in the system together so that fluctuations become correlated between molecules. This raises the possibility that a stably oscillating reaction network might need much higher numbers of molecules to effectively suppress stochasticity. To address these questions, we constructed a simplified mathematical model of the post-translational Kai oscillator based on previous studies[24,25] (Fig. 3a and Supplementary Fig. 5A). This model incorporates experimentally observed Kai protein interactions that lead to oscillatory dynamics: KaiA promotes phosphorylation of individual KaiC hexamers, and without KaiA, KaiC dephosphorylates[13]. When KaiC reaches a critical phosphorylation state, it switches into a KaiA-resistant, dephosphorylating mode[17]. Because phosphorylation is ordered[13], the sequence of states KaiC visits during the phosphorylation phase (yellow box in Fig. 3a) is distinct from the dephosphorylation phase (blue box in Fig. 3a). Finally, the dephosphorylating form of KaiC binds KaiB which then captures and inhibits KaiA, forming a delayed negative feedback loop. This mathematical model is a simplification used to study molecular noise in the basic feedback mechanism and omits known features such as subunit exchange[26]

and the localization of Kai proteins in vivo[27] which are likely to be important to understand the performance of the clock in detail.

Consistent with previous modeling work[13,17,28], the post-translational feedback loop in this model produces free-running oscillations in the deterministic limit, corresponding to infinite numbers of protein molecules (Supplementary Fig. 5B). To simulate the circadian clock at copy numbers relevant to single bacterial cells, we implemented stochastic simulations of this reaction network based on the Gillespie algorithm[29]. Similar to our experimental results, as Kai protein copy number is decreased, oscillations become noisier and the timing between cycles becomes variable (Fig. 3b, c).

Though many models of the Kai oscillator can produce circadian rhythms when the role of molecular noise is ignored, we find that different models[13,17,20] give strikingly different estimates of the magnitude of stochastic effects, implying that stochastic effects can be used to discriminate between alternative mechanisms (Supplementary Fig. 6C). We suspected that in addition to the degree of cooperativity in a model, the impact of stochasticity depends on the number of steps in the phosphorylation cycle required to switch between phosphorylation and

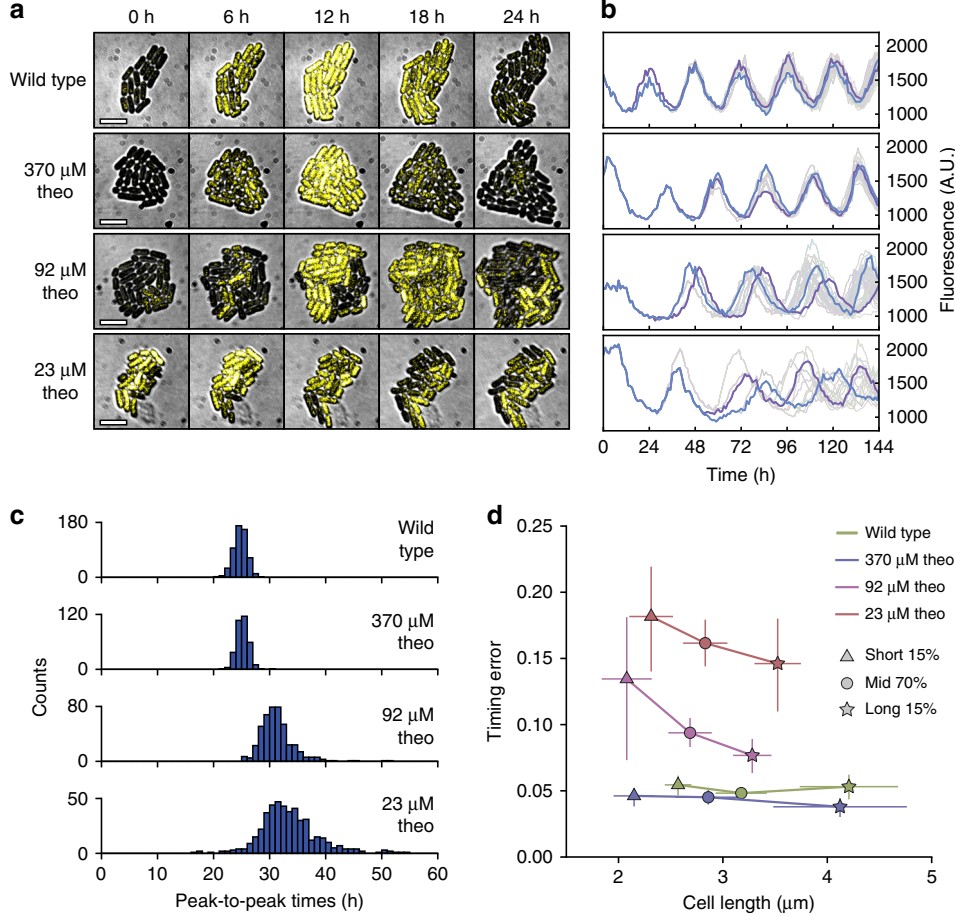

**Fig. 2** Single cell microscopy reveals desynchronized oscillations at low Kai copy number. **a** Filmstrips of YFP oscillations in wild-type cells and the copy number tunable strain induced with 1 µM IPTG and various theophylline concentrations (brightfield and YFP fluorescence overlaid). Scale bar: 5 µm. **b** Single cell oscillator trajectories (gray) with two example cell lineages highlighted (blue and purple). **c** Distributions of peak-to-peak times in wild-type and copy number tunable cells; $n = 536$ (wildtype), 336 (370 µM), 455 (92 µM), 616 (23 µM). **d** Cell length vs. timing error (standard deviation/mean of peak-to-peak intervals) in the 15% shortest cells (triangles), middle 70% cells (circles), and 15% longest cells (stars) for each condition. Vertical error bars indicate 95% confidence intervals from bootstrapping (5000 iterations), and horizontal error bars indicate standard deviation in cell length

dephosphorylation modes. Because the model we implement here allows us to vary this number of elementary steps in the reaction loop while holding other properties constant, we explored stochasticity as a function of loop size, with five steps giving the best match to the experimental data (Supplementary Fig. 6A, Fig. 3d). Previous work has shown that the transcriptional feedback loop which we have removed from our experimental strain can also work to reduce stochasticity in single cells[20]. We analyzed stochasticity in a model that includes transcriptional feedback on KaiC expression and conclude that although transcriptional feedback can reduce the effects of noise, it is unlikely to be sufficient to fully suppress the stochasticity we observe at the lowest copy numbers (Supplementary Fig. 6B).

In a reaction with many copies of the Kai proteins, we expect that stochastic fluctuations will be suppressed because the reaction averages over many molecules. Surprisingly, our results indicate that even with 1000s of Kai protein copies, timing error in the model may still be >10% per cycle. Since the behavior of each KaiC hexamer is not independent, but is coupled to the rest of the reaction by sequestration of a shared pool of KaiA[24,25], we hypothesized that stochastic fluctuations in the components of this post-translational feedback might be most responsible for timing errors in the oscillator.

To test this, we computationally injected noise into specific nodes of the reaction network to find molecular species where

molecular stochasticity caused the largest changes in oscillator phase. We found that the molecular complexes most susceptible to noise contain KaiA, precisely the molecules involved in the delayed negative feedback loop (Fig. 3e). The vulnerability of the oscillator to fluctuations in KaiA-containing complexes can be understood in terms of the sensitivity of KaiC phosphorylation rates to the amount of active KaiA. At our low inducer conditions, the number of KaiA-sequestering complexes needed to shift the entire reaction from phosphorylation to dephosphorylation is only ~10 copies (Fig. 3f). These KaiABC complexes represent only a small fraction of total KaiC (Fig. 3f) —thus the stochastic fluctuations from small numbers of KaiABC complexes can be sufficient to cause significant fluctuations in KaiC enzymatic rates. Together, these results suggest that the negative feedback loop is a dominant source of noise in the post-translational oscillator.

Although the KaiA-dependent negative feedback loop is the step most vulnerable to molecular noise, it also performs the crucial function of synchronizing individual KaiC hexamers within a single cell[24]. Left uncoupled, individual KaiC hexamers would progress through phosphorylation cycles with irregular timing, and the circadian rhythm would rapidly die out. In this way, our results suggest that the negative feedback loop is both a necessity and a liability: while it is needed to sustain free-running oscillations, Kai proteins must be expressed at ~10,000 copies per

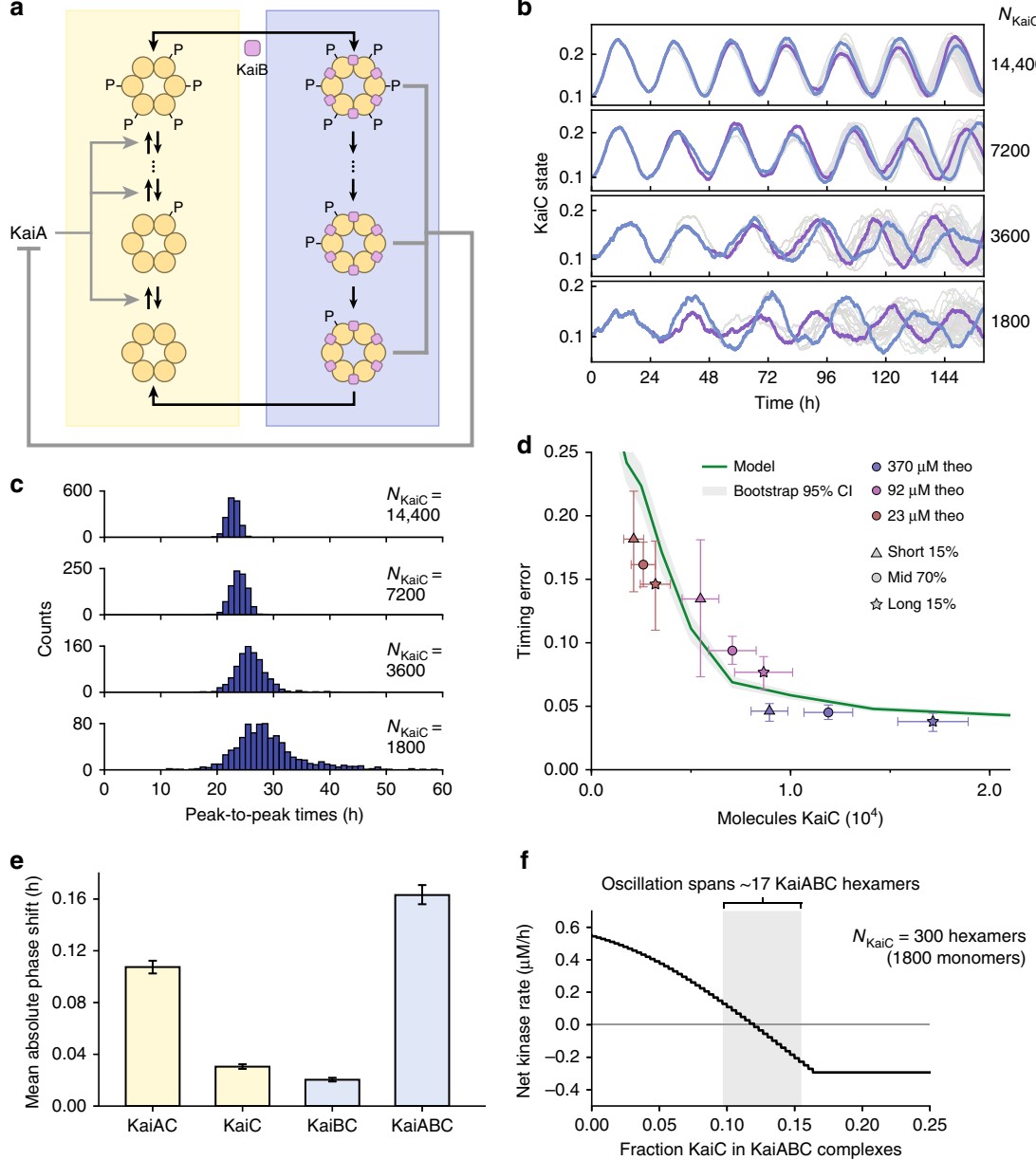

**Fig. 3** KaiA-dependent negative feedback loop is the noise bottleneck in a stochastic model of the Kai system. **a** Model of post-translational oscillator. KaiC hexamers undergo ordered phosphorylation (yellow box) and dephosphorylation (blue box). KaiA is required for KaiC phosphorylation, and dephosphorylating KaiC binds to KaiB to sequester and inhibit KaiA. (see Supplementary Fig. S5). **b** Simulated stochastic single cell trajectories (gray) at various Kai copy numbers with two example traces highlighted (blue and purple). "KaiC state" indicates the average position of KaiC molecules in the oscillator loop. **c** Distributions of peak-to-peak time intervals in the stochastic model. **d** Comparison of model and experimental data. Vertical error bars indicate 95% confidence interval from bootstrapping (5000 iterations). Horizontal error bars indicate standard error of the mean ($n = 3$). Gray interval indicates the 95% bootstrapping confidence interval for the model. **e** Mean phase shift caused by Poisson noise perturbations to molecular species in the model ($n = 500$ trials). Error bars indicate 95% confidence interval from bootstrapping. **f** Instantaneous KaiC phosphorylation rate vs. fraction of KaiC in KaiABC complexes in the stochastic model for $N_{KaiC} = 300$ hexamers. Shaded area indicates the range over which KaiABC complexes oscillate

cell to suppress the noise amplification inherent in the negative feedback loop and keep time accurately over several days.

**KaiC rhythms in *Prochlorococcus* are not self-sustaining.** The finding that high Kai protein expression is necessary to generate precise rhythms has provocative implications. Circadian rhythms are a well-known strategy that allows organisms to robustly anticipate future events using internally generated oscillations. However, our analysis suggests that there is a minimum biosynthetic investment needed to create a reliable oscillator using

the Kai proteins. Microbial cells span a wide range of sizes; for very small cells, expressing many thousands of copies of clock proteins may not be tenable. This suggests that tiny cells may use alternative dynamical strategies to keep time.

To investigate this possibility, we focused on the small cyanobacterium *Prochlorococcus marinus MED4*, whose cell volume is 10–20 times smaller than *S. elongatus*[30] (Supplementary Fig. 7 and Supplementary Note 1). Using quantitative immunoblotting, we found that *P. marinus* has ~700 copies of KaiC per cell (Fig. 4a and Supplementary Fig. 8), which is in the regime where the *S. elongatus* oscillator becomes extremely error-

prone (cf. Fig. 3d). Expressing 10,000s of Kai proteins to achieve noise suppression, as in *S. elongatus*, may not be feasible in *P. marinus* given that this investment in protein synthesis would represent ~20% of the proteome (Supplementary Note 1).

The *kaiA* gene at the heart of the negative feedback loop is missing in *P. marinus*, suggesting a qualitatively different time-keeping mechanism (previous literature[31] and Fig. 4b).

We measured KaiC phosphorylation in both light-dark cycles and constant conditions and observed that, similar to a previous report[32], the Kai system in *P. marinus* functions as an environmentally driven hourglass or timer—KaiC phosphorylation increases in the light and decreases in the dark, but, unlike a circadian rhythm, ceases to cycle when the environment is held constant (Fig. 4c, compare to Supplementary Fig. 9). The

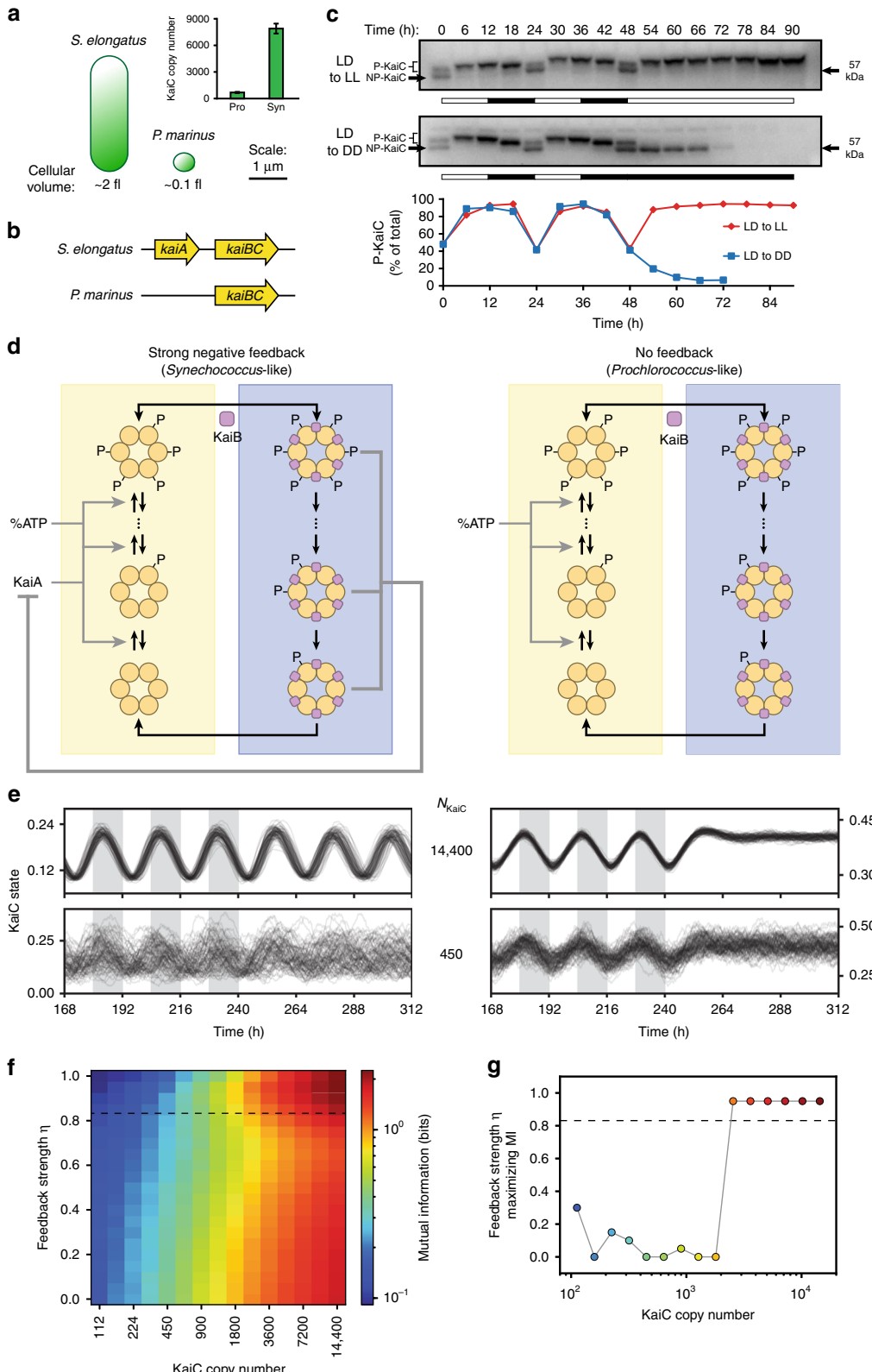

lack of self-sustained oscillations in KaiC phosphorylation is likely the molecular explanation for the lack of free-running rhythms in gene expression in this microbe[31].

**Timers can outperform true clocks if internal noise is high**. Does the alternative strategy of a driven timer without a feedback loop offer resistance to molecular noise? To computationally test this hypothesis, we extended our stochastic model of the Kai system, allowing us to vary the strength of the KaiA-dependent negative feedback loop to interpolate between a circadian rhythm and an environmentally-driven timer (Fig. 4d and Supplementary Fig. 5A). Though the mechanism of environmental input to the *P. marinus* Kai system is not known, we model the input as a modulation of KaiC phosphorylation rates, similar to the effect the ATP/ADP ratio has on KaiA-dependent KaiC phosphorylation in the *S. elongatus* system[33].

To quantify the performance of these systems, we calculated mutual information between KaiC phosphorylation and the time of day in an environment with both a regular day-night cycle and random input fluctuations simulating weather. At high copy number, mutual information is maximized by a strong negative feedback loop that produces free-running oscillations. In contrast, at low copy number, the system that maximizes mutual information has a very weak or non-existent feedback loop, corresponding to an environmentally-driven timer (Fig. 4e–g and Supplementary Fig. 10). The critical copy number below which a non-free-running timer outperforms a true circadian clock depends on the strength of random fluctuations in the input signal; increasing input noise tends to favor a stable oscillator (Supplementary Fig. 10).

## Discussion

This study reveals that the delayed negative feedback loop that sustains circadian rhythms can itself be a liability that amplifies molecular noise. Our experimental and computational analyses suggest that an alternative time-keeping strategy can be employed when protein copy numbers are low: an hourglass-like system without a feedback loop can outperform a free-running circadian clock when molecular noise is substantial. *P. marinus* apparently diverged from a common ancestor with marine *Synechococcus* that has a *kai* gene cluster with all three genes intact[31]. It has undergone both a reduction of cellular volume and a general genome streamlining that includes the loss of *kaiA*[31]. In light of our analysis, the loss of *kaiA* and the loss of stable rhythmicity may not only reflect the benefits of genome reduction, but it may also be an adaptation that outperforms a free-running clock in a low protein copy number niche.

Because *Prochlorococcus* species are typically found in near-equatorial waters of the open ocean, previous studies have suggested that the hypothesis that a free-running clock may be dispensable because of the high regularity of environmental cycles[34]. This hypothesis about the external environment is consistent with our conclusion that high levels of internal noise disfavor free-running clocks. Indeed, in our numerical simulations, free-running behavior is most favored when both internal noise is low and when the external environment has large irregular fluctuations (Supplementary Fig. 10). Thus, both of these effects may have contributed to the loss of free-running rhythms in *P. marinus*.

This result may be of broad significance to microbial physiology. The classical study of circadian rhythms focuses on oscillators that free-run in constant conditions, but our analysis suggests that for cells whose internal biochemistry is unreliable, non-free-running systems may perform better as time-keepers. This may be of particular relevance in niches with some environmental rhythmicity, such as the mammalian gut. Population oscillations have been observed in the gut microbiome[35,36], but there is currently little evidence for free-running rhythms in gut microbes themselves. By expanding our perspective away from the precise, free-running rhythms of *S. elongatus*, we may uncover a broader world of environmentally-driven timing systems in prokaryotes.

## Methods

**Cloning and strain construction**. The copy number tunable strain was constructed by transforming a *kaiABC* knockout plasmid (pJC003, gentamycin resistance) into the *kaiABC* locus of wild-type *S. elongatus* carrying either an EYFP-SsrA fluorescence reporter driven by the *kaiBC* promoter (strain MRC1006, reporter first used in previous work[37]) or the *luxABCDE* cassette driven by the *psbAI* promoter (strain MRC1005, reporter strain first used in previous work[38]), followed by transformation of a plasmid carrying the three *kai* genes and *lacI* (pJC073-2, spectinomycin resistance) into neutral site I. The two versions of the copy number tunable strain carrying the YFP reporter or luciferase reporter are denoted as MRC1139 and MRC1138, respectively.

pJC003 was constructed from a pBSK+ backbone by flanking a gentamycin resistance cassette with 300-330 bp sequences upstream of *kaiA* and downstream of *kaiC*. pJC073-2 was constructed from a pAM2314 backbone through multiple rounds of Gibson assembly. The *lacI* cassette and IPTG-inducible *trc* promoter was cloned from pAM2991. Transcription of *kaiB* and *kaiC* are driven by two separate copies of the *S. elongatus glnB* genomic promoter (750 bp upstream of the putative transcriptional start site of *glnB*). A synthetic theophylline-inducible riboswitch[15] (riboswitch F) was placed immediately upstream of the start codons of all three *kai* genes. A synthetic terminator, Bba_B0015 from the iGEM Registry of Standard Biological Parts, was placed downstream of each *kai* gene. Thirty-bp randomized linker sequences were placed upstream of the *glnB* promoter sequences for *kaiB* and *kaiC* to allow for proper Gibson assembly and plasmid sequencing.

The strain used in Supplementary Fig. 3F was constructed by transforming plasmid pMR0151 into wild-type *S. elongatus* to generate strain MRC1036. Plasmid pMR0151 was constructed by ligating *eyfp* into the BamHI and EcoRI restriction sites on pAM2991.

**Culture conditions**. *Synechococcus* cultures were grown and maintained at 30 °C in BG11 medium supplemented with 20 mM HEPES (pH 8.0) with shaking at 180 rpm under constant illumination of 75 μmol photons m$^{-2}$ s$^{-1}$, and *Prochlorococcus* cultures were grown and maintained at 22 °C in Pro99 medium[39] based on natural seawater (Woods Hole, MA) supplemented with 0.59 M NaHCO$_3$ under constant illumination of 16 μmol photons m$^{-2}$ s$^{-1}$ without shaking. Culture conditions for specific experiments are described in their respective sections.

To guard against potential genetic instability in the copy number tunable strain, all experiments were performed on cultures propagated for two weeks or less from the original freezer stock. We verified that no genomic loss of our engineered *kaiB* or *kaiC* expression system was detectable by genomic PCR or western blot in these cultures (data not shown).

**Fig. 4** Removing the negative feedback loop creates a noise-resistant environmental timer in a *Prochlorococcus*-like system. **a** Comparison of cell volume and KaiC copy number in *Prochlorococcus marinus* vs. *Synechococcus elongatus*. Copy number (inset) determined by quantitative western blot ($n = 3$). **b** *Prochlorococcus* has a simplified Kai architecture that lacks *kaiA*. **c** Top: western blot time course showing *Prochlorococcus* KaiC (ProKaiC) phosphorylation in cultures incubated in light-dark cycles followed by constant light or constant dark. Bottom: quantification of ProKaiC phosphorylation. **d** Comparisons of model architectures corresponding to *Synechococcus* (left, strong feedback) and *Prochlorococcus* (right, no feedback). **e** Simulations of Kai systems with strong feedback (left) and no feedback (right) in light-dark cycles (shaded regions), followed by constant light at high copy number (top, 14,400 KaiC copies) and low copy number (bottom, 450 KaiC copies). "KaiC state" indicates the average position of KaiC molecules in the oscillator loop. **f** Mutual information between the clock and time of day during light-dark cycles in the presence of environmental fluctuations (see Supplementary Information). Stable oscillations occur for feedback strength above 0.83 (dashed line). **g** Feedback strength that maximizes mutual information as a function of KaiC copy number. Above the dashed line, the system shows self-sustaining circadian rhythms. Marker colors correspond to the colorbar in **f**

**Time-lapse microscopy**. To prepare cells for time lapse microscopy, cultures of either wild-type cells expressing the YFP reporter (MRC1006) or the copy number tunable cells expressing the YFP reporter (MRC1139) were grown in black, opaque 96-well plates and illuminated with custom-build LED arrays powered by an Arduino, which delivered 1.33 V across each LED (627 nm wavelength), illuminating cells with $\sim$8.8 μmol photons m$^{-2}$ s$^{-1}$. The cells were seeded at an initial OD750 of 0.1 and were entrained with two 12 h:12 h light/dark cycles. Forty-eight hours after initial seeding, wells containing duplicate culture conditions were combined into single tubes.

After pooling cultures into single tubes, 1 μL of culture was pipetted into individual wells of a glass coverslip-bottomed 96-well plate (Mat-tek corporation). For each well, a BG11-agar pad (1 mm × 2 mm × 2 mm) was placed on top of each droplet of culture. 225 μL of molten BG11-agar cooled to 37 °C and containing appropriate concentrations of IPTG and theophylline was pipetted into each well and left to cool and solidify.

Time lapse microscopy was performed with an Olympus IX-71 inverted microscope with motorized stage and focus control, and automation of image acquisition was implemented with the Micromanager software package. Images were captured with a 100x Olympus oil immersion objective and a Luca EMCCD camera (Andor). The microscope was housed in a custom-built incubator that maintained temperature at 30 °C and insulated the apparatus from external light sources. The cells were exposed to a continuous light source of 2 μmol photons m$^{-2}$ s$^{-1}$ of light (660 nm wavelength), and the illumination condenser was removed in order to widen the light beam to sufficiently illuminate multiple wells evenly. Over the course of one hour, the microscope imaged 24 unique fields of view with brightfield, chlorophyll, and YFP filter sets (exc. 500 nm/20 nm bandpass, emm. 535 nm/30 nm bandpass, dichroic 515 nm long bandpass).

Microscope experiments using light-dark cycles were performed as described above with the following exceptions. Cells were entrained in the 96-well plate with two 16:8 h light/dark cycles, at which point they were transferred to the microscope. Cells were allowed to acclimate to a constant light environment for one 24-h period, at which point they were subjected to six 16:8 h light/dark cycles followed by release into constant light. Images were only acquired during the 16-h light period. Light levels during each step were the same as described above.

**Single cell image processing and data analysis**. Cell masks for image processing were obtained using custom-written Python software to allow the user to manually draw a "mask estimate" over individual cells in brightfield images. Using the estimated masks as initial guesses, the software optimized mask areas to fit the underlying cells such that the total pixel intensity was minimized, utilizing the relatively dark cell interior to do so. The fitted cell masks from one movie frame were then used as initial guesses for cells in the next frame, and any errors were corrected manually. After the cell masks were labeled for the duration of the experiment, the CellTracker software suite[40] was used to construct cell lineages based on these cell masks to measure cellular YFP fluorescence intensity over time for individual lineages.

Peak to peak intervals were detected using a Python implementation of the Matlab peak detection algorithm, and algorithm parameters were tuned to find all local maxima without any restrictions on minimum distance between maxima. All cell lineages within a single microcolony (i.e., all of the descendants of a single mother cell at the start of the experiment) share varying degrees of overlap due to common ancestry, so to avoid counting the same data multiple times, we only considered unique peak to peak intervals, defined as intervals that occurred within unique pairs of mother/daughter cells or at unique times in the experiment if the peaks occurred within the same cell. To prevent peak identification from identifying spurious peaks originating from high frequency measurement noise, the lineage data was smoothed with the Savitsky-Golay algorithm using a window size of 11 timepoints and third order polynomial fitting before peak finding. Statistics were then calculated for the peak to peak distribution mean, standard deviation, and coefficient of variation (defined as standard deviation divided by the mean). Error bars were estimated as the 95% confidence intervals from 5000 iterations of bootstrapping analysis of the experimental data. Oscillation amplitude was measured by quantifying the difference between peaks and the troughs that immediately preceded them (troughs were identified using the same peak detection algorithm described above).

**Western blotting analysis**. To prepare cells for western blotting to quantify cellular Kai copy number, cultures were grown in black 96-well plates illuminated with the Arduino-controlled LED array described above under constant illumination, and cells were not subjected to any prior entrainment protocols before growth in 96-well plates. Cultures of the copy number tunable strain were supplemented with 1 μM IPTG and varying concentrations of theophylline. The cells were seeded at an initial OD750 of 0.3 with each sample distributed across 12 wells (200 μL culture/well), and they were allowed to grow for 48 h. At this point, 1.75 mL of culture was taken per sample, pelleted at 3000 × $g$, flash frozen in liquid nitrogen, and stored at −80 °C.

To prepare *Prochlorococcus* cultures for western blotting to quantify cellular KaiC copy number, cells were seeded at an initial OD750 of ~0.003, and they were grown to a final OD750 of 0.09, at which point 22.2 mL cells per sample were pelleted at 3000 g, flash frozen in liquid nitrogen, and stored at −80 °C.

Frozen cell pellets were resuspended in lysis buffer containing 8 M urea, 20 mM HEPES pH 8.0, 1 mM MgCl$_2$, and 0.5 μL benzonase (EMD Millipore). Samples were then lysed with 10 cycles of vortex bead beating using 0.1 mm glass beads. Complete lysis was verified by microscopy (*Synechococcus*) or flow cytometry (*Prochlorococcus*). Sample protein concentration was measured by Bradford assay, using BSA as a protein standard. Samples were then mixed with 3× SDS-PAGE sample buffer (150 mM Tris-Cl pH 6.8, 6% SDS, 300 mM DTT, 30% glycerol, 0.1% bromophenol blue) and immediately loaded into polyacrylamide gels for SDS-PAGE.

For quantification of either cellular Kai expression or KaiC phosphorylation dynamics, samples were resolved in 4–20% TGX gels or 7.5% Tris-HCl gels, respectively (Biorad). Gels were transferred onto PVDF membrane (Biorad) and blocked in 2% milk + TBST (137 mM NaCl, 2.7 mM KCl, 20 mM Tris, 0.05% Tween-20, pH 7.4). Membranes were then incubated in primary antibody, washed in TBST, and incubated in secondary antibody. Antibody information is listed in (Table 1).

Membranes were visualized with SuperSignal West Femto substrate (Thermo Fisher) and imaged (Biorad ChemiDoc MP). Bands were quantified by densitometry in ImageJ against purified recombinant protein standards, and the intensities of non-specific bands (determined from *kaiABC* null samples) were subtracted. In accordance with a protocol from previous work[28], *Synechococcus* recombinant protein was prepared as described below, and *Prochlorococcus* KaiC was prepared as described below. To minimize quantification of non-Kai protein, quantification of recombinant Kai protein standards was performed by running dilution series of standards in SDS-PAGE gels against a BSA standard dilution series followed by staining and imaging with SimplyBlue SafeStain (Life Technologies), in which only the bands with the appropriate molecular weight were quantified.

**Preparation of recombinant *Synechococcus* Kai proteins**. *Synechococcus* KaiA was expressed from the pGEX-6P-1 plasmid[33] in DH5α *E. coli* as an N-terminal GST fusion protein, and overnight cultures grown fresh after transformation were used to inoculate 1 L of LB medium with 50 μg/mL carbenicillin. The 1 L culture was grown at 37 °C to an OD$_{600}$ of approximately 0.6, at which point it was cooled to 16 °C and induced overnight with 100 μM IPTG. The cells were subsequently harvested, pelleted, and flash-frozen with liquid nitrogen. The day the protein extraction was performed, cell pellets were resuspended and lysed using an Emulsiflex-C3 homogenizer (Avestin), and soluble KaiA in the lysate was trapped using a GSTrap column (GE Healthcare) and washed with buffer containing 1 mM ATP. To remove the GST tag, the column containing bound KaiA was incubated overnight at 4 °C with Precission protease (GE Healthcare). Cleaved KaiA was then passed through a Resource Q anion exchange column (GE Healthcare) and eluted in concentrated form (20–30 μM) in buffer with 10% (vol/vol) glycerol, which was flash frozen in small aliquots.

*Synechococcus* KaiB was expressed from the pET47b(+) plasmid[33] in BL-21 *E. coli* with an N-terminal His6 tag, and overnight cultures grown fresh after transformation were used to inoculate 1 L of LB medium with 50 μg/mL kanamycin. The 1 L culture was grown at 37 °C to an OD$_{600}$ of approximately 0.6, at which point it was cooled to 30 °C and induced for 3 h with 100 μM IPTG. The cells were subsequently harvested, pelleted, and flash-frozen with liquid nitrogen. The day the protein extraction was performed, cell pellets were resuspended and lysed using an Emulsiflex-C3 homogenizer (Avestin), and soluble KaiB in the lysate was bound using a HisTrap HP column (GE Healthcare). To remove the His tag, the column containing bound KaiB was incubated overnight at 4 °C with HRV-3C protease (EMD Millipore). Cleaved KaiB was then passed through a Resource Q anion exchange column (GE Healthcare) and eluted in concentrated form (~50 μM) in buffer with 10% (vol/vol) glycerol, which was flash frozen in small aliquots.

*Synechococcus* KaiC was expressed from the pRSET-B plasmid[33] in BL-21 *E. coli* with an N-terminal His6 tag, and multiple colonies obtained after overnight transformation were used to inoculate 1 L of Terrific Broth medium with 50 μg/mL carbenicillin. The 1 L culture was grown at 25 °C for 48 h without induction, in which KaiC was expressed due to the leaky T7 promoter. The cells were subsequently harvested, pelleted, and flash-frozen with liquid nitrogen. The day the protein extraction was performed, cell pellets were resuspended and lysed using an Emulsiflex-C3 homogenizer (Avestin), and soluble KaiC in the lysate was bound using a HisTrap HP column or HiTrap TALON column (GE Healthcare). To remove the His tag, the column containing bound KaiC was incubated overnight at

### Table 1 Antibody information

| Antibody target | Antibody host | Dilution | Source |
| --- | --- | --- | --- |
| KaiA | Rabbit | 1:2500 | Gift from Susan Golden lab |
| KaiB | Rabbit | 1:500 | Custom ordered (Rust lab) |
| KaiC | Rabbit | 1:5000 | Custom ordered (Rust lab) |
| anti Rb | Goat | 1:10,000 | Biorad (cat #1706515) |

  

4 °C with HRV-3C protease (EMD Millipore). Cleaved KaiC was then passed through a HiPrep 16/60 S-300 size-exclusion column (GE Healthcare). The elution fractions corresponding to the molecular weight of KaiC were pooled and concentrated to 40–55 µM using a 30-kDa centrifugal filter (Millipore) in buffer with 10% (vol/vol) glycerol, which was flash frozen in small aliquots.

**Preparation of recombinant *Prochlorococcus* KaiC.** *Prochlorococcus kaiC* was cloned from the MED4 genome with an N-terminal 6X His-tag followed by a HRV-3C protease site and inserted in between the KpnI and EcoRI restriction sites of the pMAL-c5e vector. The addition of the Maltose binding protein (MBP) tag from this vector was used to improve solubility of the KaiC protein in *E. coli* during expression. This plasmid was transformed in BL-21 *E. coli* cells and expressed at 18 °C for 48 h without induction. Cells were lysed by high pressure homogenization using an Emulsiflex homogenizer, and the lysate was clarified by centrifugation at 30,000 g for 1 h. This clarified lysate was applied through a Ni-NTA column and *Procholorococcus* KaiC (ProKaiC) was eluted with an imidazole gradient. HRV-3C protease (ThermoFisher Scientific) was added to eluted fractions containing ProKaiC to cleave off the MPB-6X His tags by incubation overnight at 4 °C. The post-cleavage fractions were concentrated and further purified via size exclusion chromatography using a Hiprep 16/60 S300 column and an elution buffer containing 150 mM NaCl, 20 mM Tris pH 8.0, and 1 mM ATP. It was determined by SDS-PAGE that the MPB-6X His tags were incompletely cleaved after the size exclusion step. Hence, fractions containing Pro-KaiC were again incubated overnight with HRV-3C at 4 °C to achieve complete tag cleavage. The added HRV-3C protease, which contained a 6X His-tag, and uncleaved ProKaiC was then removed via incubation with Ni-NTA resin. KaiC concentration in the resulting supernatant was estimated by gel densitometry. This solution was used as a recombinant protein standard for quantitative Western blotting.

**Protein copy number measurement.** Kai protein copy number per cell was quantified by determining the amount of Kai protein in cell pellet samples by quantitative western blotting (described above) and dividing by the number of cells in the pellet (Supplementary Data 1).

Synechococcus cell counts were determined by pipetting 1 µL of diluted sample into a 96-well plate and adding an agar pad and BG11-agar as described in the time lapse microscopy methods section. The microscope was then programmed to tile the entire well to image all the cells by chlorophyll autofluorescence, and the cells were then manually counted in ImageJ. *Prochlorococcus* cells were too small to count reliably on the microscope, and thus absolute cell counts were obtained with an Attune Acoustic Focusing Cytometer, which has been previously used to quantify cell counts for *Prochlorococcus*[41,42].

Reported values for Kai protein copy numbers are the averages of three independent biological replicates for each condition. Uncertainties were calculated as the standard error of the mean for each estimate.

**Electron microscopy.** Synechococcus and Prochlorococcus cells were grown and pelleted in identical conditions used for western blotting, described above. Pelleted cells were fixed with 2% glutaraldehyde and 4% paraformaldehyde in 0.1 M sodium cacodylate buffer (Electron Microscopy Sciences) overnight at 4 °C. Cells were then washed with sodium cacodylate buffer. The buffer was removed and replaced with 1% osmium tetroxide in 0.1 M sodium cacodylate buffer for 60 min, after which cells were washed again with sodium cacodylate buffer. Cells were then washed with maleate buffer, pH 5.1 (50 mM maleic acid titrated with NaOH), 1% uranyl acetate in maleate buffer, followed by maleate buffer. The cells were then dehydrated using serial washes of ethanol (25, 50, 70, 95%, then 100%) followed by 100% propylene oxide. Cells were embedded into resin by incubation in 2:1 propylene oxide:Spurr resin, 1:1 propylene oxide:Spurr resin, then 100% Spurr resin, after which the resin was polymerized by incubation at 60 °C for 1–2 days.

Ninety nanometer slices were sectioned (Reichert-Jung Ultracut E Microtome) and were stained with uranyl acetate and lead citrate. Images were acquired at 300 kV on a FEI Tecnai F30 transmission electron microscope with a Gatan CCD camera.

**Circadian bioluminescence measurements.** Bioluminescence measurements of either the wild-type strain carrying a luciferase reporter (MRC1005) or the copy number tunable strain carrying a luciferase reporter (MRC1138) were obtained using a PerkinElmer TopCount Microplate Scintillation and Luminescence Counter. Black, opaque 96-well plates were prepared by pipetting 250 µL of BG11-agar into each well. After the agar solidified, 10 µL of 25× inducer at varying concentrations (IPTG + theophylline) was pipetted onto the top of the agar, and the plate was left overnight to allow the inducer to diffuse uniformly throughout the agar. Cells were pipetted onto the plate and illuminated with the Arduino-controlled LED arrays as described above, and they were subjected to two 12 h:12 h light/dark cycles followed by release into constant light, at which point the plate reader began taking bioluminescence measurements every 30 min. Data curves shown for each condition and strain represent the average of data recorded from four replicate wells.

To quantify the period and amplitude of circadian oscillations, the data were first detrended by dividing by a best fit line for the duration of the experiment after

release into constant light (111 h total). This detrended data was then fit to a sinusoid.

**In vitro measurement of KaiABC oscillations using a fluorescent probe.** KaiA, KaiB, KaiC were prepared according to standard protocols[43], but the anion exchange chromatography step was omitted in the KaiC purification protocol. Fluorescently labeled KaiB (KaiB-IAF) was prepared by an overnight labeling reaction of KaiB-K25C with 6-iodoacetofluorescein[43]. Protein concentrations were measured via densitometry on an SDS-PAGE gel against a BSA standard (Biorad). KaiABC reactions at different concentration were prepared by serial dilution of a master mix prepared at 2× standard concentration (7 µM KaiB and KaiC, 3 µM KaiA, 0.4 µM KaiB-IAF) in standard reaction buffer containing 2.5 mM ATP (20 mM Tris [pH 8], 150 mM NaCl, 5 mM MgCl2, 10% glycerol, 0.5 mM EDTA, 2.5 mM ATP). To monitor oscillations, reactions were transferred to a black 384-well plate (25 µL/well) and placed into a Tecan F500 plate reader pre-warmed to 30 °C[43]. Parallel and perpendicular fluorescence signals (exc. 485 nm, 20 nm bandpass; emm. 535 nm, 25 nm bandpass; dichroic 510 nm.) were read out every 15 min. Wells containing buffer only were used as blanks. The G factor was adjusted to give 20 mP in the wells containing a solution of free fluorescein.

**Code availability.** The source code used for custom microscope image analysis, including generation of cell masks, is available at https://github.com/jwchew/rustlab_microscopy_analysis.

**Data availability.** Data not presented within this manuscript is available from the authors by request.

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

## Acknowledgements

We thank Connie Phong for assistance with purified proteins, Guillaume Lambert for assistance with microscopy, and Ed Munro, Joe Markson, and the Rust Lab for useful discussions. Maureen Coleman and Jesse Black provided assistance with flow cytometry and culturing *Prochlorococcus*. In addition, we thank Susan Golden for a generous gift of KaiA antibody and the Open Science Data Cloud for free usage of computational resources. This work was supported by NIH training grant T32-GM007281, NIH F30 fellowship F30-GM117962 (to J.C.), NIH R01-GM107369 and a Pew Biomedical Scholars award (to M.J.R.).

## Author contributions

J.C. and M.J.R. designed experiments and prepared the manuscript. J.C. carried out experiments and data analysis, E.L. performed fluorescent measurements of in vitro Kai oscillations, and J.L. designed and performed the protocol for purifying *Prochlorococcus* KaiC and edited the manuscript. J.C., M.J.R. and A.M. designed and analyzed the mathematical model, and J.C. implemented the model and analyzed output data.

## Additional information

**Competing interests:** The authors declare no competing interests.

