## [Peer Review File · Nature Communications]

Reviewers' comments:

Reviewer #1 (Remarks to the Author):

The authors have combined an impressive single-cell analysis of two very different cyanobacterial species, *S. elongatus* PCC 7942 and *Prochlorococcus marinus*, along with stochastic modeling of the cyanobacterial Kai protein clock and Western blotting analysis to provide a valuable hypothesis of the stochasticity of the Kai system providing implications not only for Cyanobacteria but Bacteria in general. Their work is important because the functional role of the Kai clock is only well understood in the model strain *S. elongatus* PCC 7942 where it is encoded as a single cluster of *kaiA*, *kaiB*, and *kaiC*. In contrast, the Kai system likely provides alternative functions in other strains/bacteria which are likely to be industrially, clinically and ecologically important.

Major concerns:

Stochastic modeling and the resulting hypothesis suggested by the authors heavily depend on a correct absolute Kai protein quantification, wherefore the authors chose a Western blotting analysis: "Using quantitative western blotting, we found that wild type cells express ~4,000 KaiA, 51 ~11,000 KaiB, and ~8,000 KaiC copies per cell—a stoichiometry similar to that needed to support oscillations in vitro." Using Western/immune blotting, correct absolute numbers are difficult to achieve. However, such a quantification has been done before by Kitayama et al., 2003, EMBO: "In contrast, KaiA levels were almost constant and relatively low, at 5% of the lowest levels of KaiB and KaiC (Figure 1B)." This previous work should be cited and discussed, because numbers for KaiA differ by a factor of 10 at least. However, I would highly recommend a protein quantification method, which is more precise and up to date, e.g. mass spectrometry, for Kai protein quantification. Thus, one would have the chance to obtain not only absolute numbers for KaiA, KaiB and KaiC but also for KaiC-P.

The conclusions drawn from the brilliant experimental data have to be discussed further.

Stoichiometry is key to the Kai system, especially the KaiA : KaiBC ratio, which has been shown multiple times before, e.g. Nakajima et al., 2010, FEBS letters. As soon as different, in particular inducible promoters are used, numbers change between Kai proteins. Thus, asynchrony can be achieved quite easily. Further, synchronization of the Kai clock is not only given by KaiA sequestration but also by KaiC monomer exchange, which strongly depends on the 'daytime' of the system. Finally yet importantly, a spatial distribution of the Kai proteins has been suggested, e.g. Cohen et al., 2014, Curr Biol, which most likely affects the local concentration and thus stochasticity might be considered more carefully and likely spatially. One might discuss if protein copy numbers and cell size really matter if the clock ticks at the cell pole only, for example.

In summary, I would recommend changing the title of the manuscript. The statement that only protein copy number decides between timers or clocks might be misleading. Marine *Synechococcus* strains can be as small as *Prochlorococcus* cells but having the full set of *kaiA*, *kaiB*, and *kaiC* genes and a true clock, e.g. Sweeney & Borgese, 1989, J Phycol. The authors might explain/justify further why the assumption of constant protein concentrations over cell sizes is valid. Using a flow cytometer, one might be able to correlate cell size and protein concentration approximated by the corresponding reporter fluorescence.

Minor:

- 1) One might cite and discuss the theory of the deterministic limits of stochastic reaction networks; specifically also wrt. (1), the role of concentration vs. amount; wouldn't the smaller cells require less proteins for still allowing for deterministic reaction networks?
- 2) The authors used a Custom Python Code for single cell image processing and data analysis, which should be published somewhere.
- 3) "Quantification of recombinant standards was performed by running dilution series of standards in SDS-PAGE gels against a BSA standard dilution series followed by staining and imaging with SimplyBlue SafeStain (Life Technologies)." Here, photometrical methods like Bradford staining or the DirectDetect system (Merck) might be more suitable to derive at quantitative protein

numbers.

Reviewer #2 (Remarks to the Author):

In this elegant study, the authors examine the fidelity of the circadian clock in *S. elongatus*, by modulating the expression levels of the core clock proteins and examining the robustness of the resulting rhythms at the single cell level. They link this work to a stochastic model of the system, showing that high protein numbers are required for robust oscillations. They then go onto claim that such high copy numbers would not be possible in smaller cyanobacteria, such as *Prochlorococcus*. They show that *Prochlorococcus* is not capable of free-running circadian rhythms, and use modelling to suggest that an evolutionary choice is made between high copy number feedback clocks that are able to anticipate daily cues but are energetically costly, and low copy number timers that can entrain to light dark cycles, but can't anticipate daily cues. Overall the study is nicely done, can be considered for publication in *Nature Communications* and is a great read. However, I have the following comments about the paper that need addressing:

1. The transcriptional feedback loop (as KaiC indirectly modulates its own transcription) is removed in this study, as the key clock components are expressed from inducible promoters. The transcriptional feedback loop has been proposed to play a role in the robustness of the clock (Teng et al, *Science*, 2013). How would the transcriptional feedback loop affect the conclusions of the paper? This needs to be discussed, preferably with modelling results explaining how transcriptional loops affect the copy numbers required for robust rhythms. Could transcriptional feedback loop allow a small bacterium to have a robust clock? I realize it may be tricky to incorporate this loop into the Gillespie algorithm model, as it would include unknown and non-elementary reactions, so either a different stochastic implementation, or a modified Gillespie model with periodic KaiC-dependent updates on propensities/copy numbers will do.
2. The strength of the evolutionary claim that low copy number clocks perform badly under light dark cycles could be improved. Currently this is explored computationally. I would have thought that the results in Figure 4 would depend on the strength of the resetting cue in the model. If the effect of light on the clock is to cause a strong resetting of the clock to a particular phase, could low copy number clocks with feedback behave as well as timers? This is something easy to test in the model. An additional experiment that would strengthen the evolutionary claim would be to examine how the low copy number clock (ie induced at 23uM) behaves under light dark cycles in the scope. Do the rhythms look noisy as predicted? This could greatly strengthen the argument that low copy number clocks are noisy under light dark cycles.
3. In Extended data 2, why is optimal rhythmicity obtained for low levels of IPTG (even up to no induction at all)? Combined with the maximum efficiency of the theophylline expression system (25% as per the first paragraph in supplementary methods), wouldn't this essentially keep KaiA at low levels of expression, whereas 4,000 copies are measured in WT (line 50)? Is the P_trc promoter very leaky in cyanobacteria?

Minor points:

1. Is a "timer" a standard definition in the field for a non-free running clock? What is it timing? My impression by looking at figures 4C and 4E is the *Prochlorococcus* Kai system simply responds to a signal switching by moving to a different steady state, either saturating (in light) or going to a basal level in dark, as you'd expect from the dynamics of a phosphorylation-dephosphorylation system. I suppose the time-scale of these dynamics could be timing something, but it's not obvious it is timing a day or half a day in the experimental data. So is "timer" a correct terminology?
2. Is there a typo in line 68, where it is stated that 390 uM theophylline was used, whereas it is 370 in the figure?
3. In Figure 2C, the mean peak-to-peak time changes by about 30% in low copy number conditions. I found this quite interesting. Could the authors briefly comment on this effect?
4. On fitting timing error to KaiC levels while varying the number of phosphorylation steps (line

105 and Extended data figure 7): presumably there can be up to 12 steps because there are two phosphosites in each KaiC monomer. Can the authors make this clearer for those not familiar with the system, as at first it is slightly confusing that Figure 3 shows what appear to be a maximum of 6 steps? Also, while the y-axis in the supplementary figure is labeled as coefficient of variation, the same quantity is called timing error elsewhere. Perhaps it would be better to maintain consistency.

5. Can the authors expand a bit more on the necessity of including the C* state in their model (extended data figure 5)? Couldn't the ground state itself undergo non-KaiA mediated phosphorylation at an appropriately lower rate?

6. In Supplementary methods, under "Determining model timing error", the ~24h doubling time is referenced as "data not shown", which is to be avoided as per Nature Comm guidelines. No further figure is needed in my opinion, the authors could just delete that passage.

7. Line 219 – Spelling mistake - YPF rather than YFP.

Reviewer #3 (Remarks to the Author):

The manuscript, "Demand for high protein copy number can favor timers over clocks in bacteria," by Rust et al. introduces a clever hypothesis for the evolution of oscillator vs. hourglass timer mechanisms for biological timekeeping in cyanobacteria. As I understand it, the authors argue that cyanobacteria start out with a kaiABC oscillator system, but as some species (esp. *Prochlorococcus*) pursue a streamlining strategy that includes smaller and smaller cell sizes, at some point the number of Kai proteins becomes too small to support phase coherent oscillations and downgrades to a timer system, at which point KaiA is no longer needed and is (mostly) deleted from the genome. This is an imaginative idea that provides a potential selective pressure for the loss of kaiA from the *Prochlorococcus* genome, in contrast to the other scenario that has been proposed (Axmann et al. *J. Bacteriol.* 2009; Johnson et al., *Nature Reviews Microbiol.* 2017), namely that KaiA was lost from the *Prochlorococcus* genome as part of genome streamlining so that a timer resulted, and that in the consistent & regular environment of the ocean, a timer was not a liability and therefore was not selected against.

Having said that, there are a number of defects with the current manuscript that lessen the attractiveness of the study. To warrant publication in *Nature Communications*, the authors need to perform new experiments and undertake a major revision of the current manuscript.

The first part of the paper pursues a strategy in *S. elongatus* of reducing kai protein levels and finding a threshold for kai protein levels that oscillates and comparing this with the levels measured in the cells. Three items regarding this part of the paper:

1. If the data of Figures 1-3 were the only experimental support for the overall hypothesis, it would not suffice to make the point of stochastically limiting number of molecules. The data of Figures 1-3 can be interpreted merely as a concentration effect, and we know well that the in vitro oscillation will also fail at low protein CONCENTRATION when the NUMBER of molecules is not limiting. Therefore, to make their point of limiting NUMBER of molecules, the *Prochlorococcus* case is necessary to be persuasive that the effects observed by the authors are not merely due to concentration. See below for concerns about the *Prochlorococcus* data.

2. The methodology for independently controlling KaiA and KaiBC levels is imaginative and will certainly be useful for other studies in a variety of labs, so the authors are to be congratulated for designing this expression methodology.

3. The measurements for KaiA concentration differ from those published by Kitayama et al. in 2003 (KaiB and KaiC concentrations are similar). More information is needed here as this is a crucial point for their interpretations and their measurements are different from those of Kitayama and co-authors. In particular, please provide on Extended Data Figure 1 the following: panels A and C should label the amount of the purified Kai protein loaded on each lane so we don't have to guess at the intermediate amounts, and show the plots of the standard curves with error bars for

the replicate measurements. Moreover, the authors do not refer to the Kitayama et al. paper—why not? It is fundamental scientific courtesy to refer to the prior measurements, and surely the authors know about that important publication. Finally, the authors should address why their results differ from the results of Kitayama and coworkers for WT cells.

Moreover, measurements of protein abundance need to be performed under the same metabolic conditions as the rhythm analyses, i.e., agar not liquid cultures. Metabolic state varies dramatically in cyanobacteria under agar vs. liquid cultures.

Regarding the data obtained with *Prochlorococcus* cells, the authors MUST specify which strain of *Prochlorococcus marinus* they are studying. There is a large range of sizes among different *Prochlorococcus* strains (Cermak et al. ISME J. 11: 825-828, 2007), and in some cases there is only a small difference between the volumes of *Prochlorococcus* cells and *Synechococcus* cells (e.g., Bertilsson et al. Limnol. Oceanogr., 48: 1721–1731, 2003). The authors should show actual micrographs of the cells they are comparing as a helpful addition to the diagrams in Figure 4A.

Regarding the model, three concerns arise:

1. Why does the KaiC dephosphorylate in the dark in *Prochlorococcus*? Is it because of declining ATP:ADP in the dark? If so, the authors need to measure [ATP], [ADP], and [AMP] in the light & dark in *Prochlorococcus*, and express it as energy charge, as done by Puszynska & O'Shea (Elife 6: e23210, 2017). Also in that context, Puszynska & O'Shea did NOT find a significant change in energy charge in darkness in *Synechococcus*, in apparent contradiction to the previous ATP:ADP results of the current authors. How do the authors reconcile the results of Puszynska & O'Shea with their own previous results and with the current model?
2. The authors use a "simplified" mathematical model in the presentation of Figure 4. However, the senior author is well known for his modeling of the clock mechanism of cyanobacteria on the basis of both phosphorylation sites, etc. Why did the authors revert to a "simplified" model for this paper, when the authors could have easily use the more complicated model and refer to their previous papers? Does the more accurate and complicated model NOT show the effects depicted in Figure 4? I would like to see the essential simulation results repeated with the more accurate model and shown in the Supplement.
3. In the context of issue # 2 above, the authors' hypothesis crucially depends upon the enhancement of phase coherence with a timer at low copy number, as shown in Figure 4E. And indeed, the timer does better at low copy number ("450" NkaiC). However, the timer also appears to have better phase coherence in constant light at HIGH copy number ("14,400" NkaiC). This should not be the case if the negative feedback system is operating appropriately, and this comparison of the constant light simulations suggest that the model has been over-tweaked to obtain the desired result at low copy number.

Prochlorococcus KaiC phosphorylation oscillates in light/dark but not in constant conditions, thereby exhibiting what appears to be an hourglass timer. It is the hypothesis of this manuscript that an hourglass timer was selected because the number of Kai molecules became too small. But the alternative hypothesis suggested by other labs is that KaiA was lost, creating a timer and it didn't matter to the *Prochlorococcus* in the constant marine environment. In this context:

1. knowing if a kaiA-deletion strain of *S. elongatus* exhibits a KaiC phosphorylation rhythm in light/dark but not in constant light could enhance the authors' arguments.
2. the authors should acknowledge the previous discussions of the alternative model (papers mentioned above in the first paragraph of this review), and discuss the pros and cons of each evolutionary scenario.

A few minor points that should be simple to correct:

1. Lines 81-83: the authors should clarify that timing error of small cells increases at LOW induction levels. Their point is confusing as currently written.
2. Fig. 2A and accompanying movies: from the movies, it appears that the loss of phase coherence appears to occur early in the growth of the micro-colonies and appears to stabilize thereafter.

From the authors' analyses, is this observation true? And if so, how do the authors explain this factor in their overall interpretations?

3. Fig. 3B: how is the %phosphorylation calculated? Why is it so low? These are not physiological values, and may indicate a problem with the model.

4. Fig. 4E: same concerns about %phosphorylation as in #3 above (for Fig. 3B).

5. A parameter set for the numerical simulations is not given in the supplement.

6. On lines 120 & 121, the authors claim that reference # 18 supports their postulate that the KaiABC complexes represent only a small fraction of total KaiC, but in fact the data in that paper does NOT support this interpretation (Figure 2 of Kageyama et al. 2006 may appear to support the claim, but Figure 3 of that paper clearly indicates that the KaiABC complex is a large proportion of the total KaiC pool at incubation time 36).

7. In Extended Data Figure 9, it is difficult to see how the authors could have derived the clear oscillations of %P-KaiC densitometry plots from the fuzzy immunoblots in which the bands of phospho- vs. non-phospho-KaiC are not resolved well.

Reviewer #1 (Remarks to the Author):

The authors have combined an impressive single-cell analysis of two very different cyanobacterial species, *S. elongatus* PCC 7942 and *Prochlorococcus marinus*, along with stochastic modeling of the cyanobacterial Kai protein clock and Western blotting analysis to provide a valuable hypothesis of the stochasticity of the Kai system providing implications not only for Cyanobacteria but Bacteria in general. Their work is important because the functional role of the Kai clock is only well understood in the model strain *S. elongatus* PCC 7942 where it is encoded as a single cluster of *kaiA*, *kaiB*, and *kaiC*. In contrast, the Kai system likely provides alternative functions in other strains/bacteria which are likely to be industrially, clinically and ecologically important.

Major concerns:

Stochastic modeling and the resulting hypothesis suggested by the authors heavily depend on a correct absolute Kai protein quantification, wherefore the authors chose a Western blotting analysis: "Using quantitative western blotting, we found that wild type cells express ~4,000 KaiA, 51 ~11,000 KaiB, and ~8,000 KaiC copies per cell—a stoichiometry similar to that needed to support oscillations in vitro." Using Western/immune blotting, correct absolute numbers are difficult to achieve. However, such a quantification has been done before by Kitayama et al., 2003, EMBO: "In contrast, KaiA levels were almost constant and relatively low, at 5% of the lowest levels of KaiB and KaiC (Figure 1B)." This previous work should be cited and discussed, because numbers for KaiA differ by a factor of 10 at least. However, I would highly recommend a protein quantification method, which is more precise and up to date, e.g. mass spectrometry, for Kai protein quantification. Thus, one would have the chance to obtain not only absolute numbers for KaiA, KaiB and KaiC but also for KaiC-P.

We thank the reviewer for raising this issue. It was an oversight to not include a citation to Kitayama et al. 2003 in the original version of the manuscript. We have updated the text to include the following discussion of our measurements in relation to the Kitayama results:

Our estimates for KaiB and KaiC are similar to a previous report, though our estimate for KaiA is markedly higher. The stoichiometry we observe here is similar to that needed to support oscillations with purified proteins¹³

Indeed, quantitative mass spectrometry could be used to determine protein concentrations in lysate by comparing with isotopically labeled peptide standards. However, we believe that protein concentration measurement is not the dominant source of error here, rather it is variability in the total number of lysed cells in each sample. Thus, we have chosen not to pursue the mass spectrometry approach.

The conclusions drawn from the brilliant experimental data have to be discussed further. Stoichiometry is key to the Kai system, especially the KaiA : KaiBC ratio, which has been shown multiple times before, e.g. Nakajima et al., 2010, FEBS letters. As soon as different, in particular inducible promoters are used, numbers change between Kai proteins. Thus, asynchrony can be achieved quite easily. Further, synchronization of the Kai clock is not only given by KaiA sequestration but also by KaiC monomer exchange, which strongly depends on the 'daytime' of the system. Finally yet importantly, a spatial distribution of the Kai proteins has been suggested, e.g. Cohen et al., 2014, Curr Biol, which most likely affects the local concentration and thus stochasticity might be considered more carefully and likely spatially. One might discuss if protein copy numbers and cell size really matter if the clock ticks at the cell pole only, for example.

*Indeed we do find that the KaiA / KaiC ratio is critical for oscillator function, in agreement with the in vivo and in vitro studies the reviewer mentions. When this ratio is too high, oscillations fail, even at high protein copy number (shown in Extended Data figure S2). This was the motivation for designing the expression system with different transcriptional control of the *kaiA* gene vs. *kaiB* and *kaiC*. We have modified the text to include more discussion of this issue and citation of the relevant studies:*

In vitro, the ratio of KaiA to KaiC must be kept within a specific range for oscillations to occur. Thus, in our engineered strain, *kaiB* and *kaiC* are transcribed from a constitutive promoter and *kaiA* from an IPTG-inducible promoter to allow independent control of KaiA expression (Fig. 1A).

Mathematical models of the Kai oscillator (including ours) generally assume that the proteins are well-mixed on the timescale of the oscillation. We agree with the reviewer that this may not be true, based on the spatial localization results the reviewer mentioned. Though the functional role of spatial localization is not clear, it may be involved in clock stochasticity. We've added a sentence to the discussion mentioning these results:

This mathematical model is a simplification used to study molecular noise in the basic feedback mechanism and omits known features such as subunit exchange and the localization of Kai proteins in vivo which are likely to be important to understand the performance of the clock in detail.

In summary, I would recommend changing the title of the manuscript. The statement that only protein copy number decides between timers or clocks might be misleading. Marine *Synechococcus* strains can be as small as *Prochlorococcus* cells but having the full set of *kaiA*, *kaiB*, and *kaiC* genes and a true clock, e.g. Sweeney & Borgese, 1989, *J Phycol.* The authors might explain/justify further why the assumption of constant protein concentrations over cell sizes is valid. Using a flow cytometer, one might be able to correlate cell size and protein concentration approximated by the corresponding reporter fluorescence.

We agree with the reviewer that the original title may have given the misleading impression that protein copy number is the sole factor determining whether true circadian rhythms will be found in a given bacterial species. We have thus changed the title to "High protein copy number is required to suppress stochasticity in the cyanobacterial circadian clock".

To validate the previous assumption of constant protein concentration over different cell size, we provide additional experimental evidence using fluorescence microscopy showing that the average level of constitutively expressed YFP remains constant over the entire range of cell sizes observed in the single cell oscillation experiments (Fig. 2D). This quantification of YFP concentration for cells of different sizes is shown in new Extended Data Fig 3F.

Minor:

1) One might cite and discuss the theory of the deterministic limits of stochastic reaction networks; specifically also wrt. (1), the role of concentration vs. amount; wouldn't the smaller cells require less proteins for still allowing for deterministic reaction networks?

To clarify the role that molecule copy number plays in stochasticity of timing, we have added more description of the stochastic behavior of various models of the Kai post-translational oscillator, along with discussion of these numerical results in the context of the general theory of stochastic reaction networks:

We first consider a simple result from the theory of stochastic chemical reactions: if a population of N identical molecules transit independently through a sequence of m identical reaction steps, we expect the time at which half of the molecules have completed the reaction to show timing errors proportional to $1/(m\sqrt{N})$ where the factor of \sqrt{N} comes from averaging out uncorrelated fluctuations in timing between different molecules. In this case, protein copy numbers in the 1,000s would be sufficient to achieve timing errors less than 5%. However, the above scenario does not describe a self-sustaining circadian oscillator; to achieve stable oscillations feedback loops must couple the reactions of the molecules in the system together so that fluctuations become correlated between molecules. This raises the possibility that a stably oscillating reaction network might need much higher numbers of molecules to effectively suppress stochasticity.

2) The authors used a Custom Python Code for single cell image processing and data analysis, which should be published somewhere.

We have deposited the Python code used to analyze single cell fluorescence trajectories on github at https://github.com/jwchew/rustlab_microscopy_analysis (a reference for this is added in the main text).

3) "Quantification of recombinant standards was performed by running dilution series of standards in SDS-PAGE gels against a BSA standard dilution series followed by staining and imaging with SimplyBlue SafeStain (Life Technologies)." Here, photometrical methods like Bradford staining or the DirectDetect system (Merck) might be more suitable to derive at quantitative protein numbers.

All of the protein quantification measurements mentioned by the reviewer have advantages and disadvantages. We went with a gel densitometry based approach because this also allows assessment of purity. We measured protein

staining intensity in triplicate against a BSA dilution series. We cannot rule out that the absolute concentration may be slightly distorted by differential binding of the Coomassie reagent in SimplyBlue to Kai proteins vs. BSA.

Reviewer #2 (Remarks to the Author):

In this elegant study, the authors examine the fidelity of the circadian clock in *S. elongatus*, by modulating the expression levels of the core clock proteins and examining the robustness of the resulting rhythms at the single cell level. They link this work to a stochastic model of the system, showing that high protein numbers are required for robust oscillations. They then go onto claim that such high copy numbers would not be possible in smaller cyanobacteria, such as *Prochlorococcus*. They show that *Prochlorococcus* is not capable of free-running circadian rhythms, and use modelling to suggest that an evolutionary choice is made between high copy number feedback clocks that are able to anticipate daily cues but are energetically costly, and low copy number timers that can entrain to light dark cycles, but can't anticipate daily cues. Overall the study is nicely done, can be considered for publication in *Nature Communications* and is a great read. However, I have the following comments about the paper that need addressing:

1. The transcriptional feedback loop (as KaiC indirectly modulates its own transcription) is removed in this study, as the key clock components are expressed from inducible promoters. The transcriptional feedback loop has been proposed to play a role in the robustness of the clock (Teng et al, *Science*, 2013). How would the transcriptional feedback loop affect the conclusions of the paper? This needs to be discussed, preferably with modelling results explaining how transcriptional loops affect the copy numbers required for robust rhythms. Could transcriptional feedback loop allow a small bacterium to have a robust clock? I realize it may be tricky to incorporate this loop into the Gillespie algorithm model, as it would include unknown and non-elementary reactions, so either a different stochastic implementation, or a modified Gillespie model with periodic KaiC-dependent updates on propensities/copy numbers will do.

We thank the reviewer for raising this issue. Indeed, the Teng et al. study shows that removing transcriptional feedback by placing kaiBC under a constitutive promoter results in less robust oscillations. We note that when our strain is induced at the highest theophylline concentrations (370 μ M), we obtain rhythms that are comparably deterministic to the wildtype, less stochastic than those reported in the Teng et al. strain. Put another way, transcriptional feedback is not absolutely required for deterministic oscillations when copy numbers are high (consistent with the in vitro reconstitution).

To address the question of whether transcriptional feedback could cause the Kai system to function robustly at Prochlorococcus-like copy numbers, we implemented stochastic models from the Teng et. al study. Although the presence of transcriptional feedback reduces the total amount of oscillatory variability, oscillations are still predicted to be significantly noisier than wildtype Synechococcus at a Prochlorococcus-like copy number (compare Extended Data Fig 6B-C with Fig 2D). It is important to note that our stochastic implementation of the Teng et al model remains noisier than our experimental data (compare black line representing Teng et al without transcriptional feedback to experiments in Extended Data Fig 6C), so it is possible that the amount of noise is overestimated in the Teng model. However, the experimental results discussed in the above paragraph suggest that while transcriptional feedback can reduce oscillatory variability to some degree, it is unlikely that feedback alone is enough to overcome the huge increase in noise at Prochlorococcus-like copy numbers.

We have added the following comment to the main text:

Previous work has shown that the transcriptional feedback loop which we have removed from our experimental strain can also work to reduce stochasticity in single cells. We analyzed stochasticity in a model that include transcriptional feedback on KaiC expression and conclude that though transcriptional feedback can reduce the effects of noise, it is unlikely to be sufficient to fully suppress the stochasticity we observe at the lowest copy numbers (Extended Data Fig. 6B).

2. The strength of the evolutionary claim that low copy number clocks perform badly under light dark cycles could be improved. Currently this is explored computationally. I would have thought that the results in Figure 4 would depend on the strength of the resetting cue in the model. If the effect of light on the clock is to cause a strong resetting of the clock to a particular phase, could low copy number clocks with feedback behave as well as timers? This is something easy to test in the model. An additional experiment that would strengthen the evolutionary claim would be to examine how the low copy number clock (ie induced at 23 μ M) behaves under light dark cycles in the scope. Do the rhythms look noisy as predicted? This could greatly strengthen the argument that low copy number clocks are noisy under light dark cycles.

Thanks for this comment. First, the reviewer's intuition is correct: the point at which clocks become superior to timers depends on the strength of random fluctuations in the input to the system. When the environment is perfectly regular, there is no discernible disadvantage to a timer-like system. We have added an Extended Data figure (Extended Data Fig 10) that show the results of simulations analogous to those in Figure 4 with varying external fluctuation strengths. We have also added the following text to help clarify:

The critical copy number below which a non-oscillating timer outperforms a true circadian clock depends on the strength of random fluctuations in the input signal; increasing input noise tends to favor a stable oscillator (Extended Data Fig. 10).

We have also conducted single cell microscopy experiments using light-dark cycles (LD 16:8) to query the function of the Kai system at different copy numbers under driven conditions. We find that, in general, the entraining cues reduce the variance in phase compared to constant light conditions but are not sufficient to negate the effects of stochasticity. This is especially apparent at the lowest theophylline induction levels, as the reviewer surmised. Thus, we conclude that even when driven by light-dark cycles, the low copy number Kai system tracks the time of day more poorly than the high copy version. These results are presented in new Extended Data Fig 3D-E. We have added the following line to the text:

Notably, even when grown in entraining light-dark cycles, cells at low theophylline conditions show evident stochasticity in oscillator phase (Extended Data Fig. 3).

3. In Extended data 2, why is optimal rhythmicity obtained for low levels of IPTG (even up to no induction at all)? Combined with the maximum efficiency of the theophylline expression system (25% as per the first paragraph in supplementary methods), wouldn't this essentially keep KaiA at low levels of expression, whereas 4,000 copies are measured in WT (line 50)? Is the P_{trc} promoter very leaky in cyanobacteria?

Indeed, the trc promoter is leaky in cyanobacteria. With only 1 μ M IPTG, we estimate KaiA copy number to be ~580, ~920, and ~2300 at 23 μ M, 92 μ M, and 370 μ M theophylline, based on the data shown in Fig 1C.

Minor points:

1. Is a "timer" a standard definition in the field for a non-free running clock? What is it timing? My impression by looking at figures 4C and 4E is the Prochlorococcus Kai system simply responds to a signal switching by moving to a different steady state, either saturating (in light) or going to a basal level in dark, as you'd expect from the dynamics of a phosphorylation-dephosphorylation system. I suppose the time-scale of these dynamics could be timing something, but it's not obvious it is timing a day or half a day in the experimental data. So is "timer" a correct terminology?

We are using the "timer" language here in the sense of the Rensing et al. 2001 where it is used interchangeably with "hourglass clock". We agree that there is something a bit jargon-like with this usage, and we have removed this terminology from the title of the paper. Stated more plainly, we observe a slow time constant for the response of KaiC phosphorylation to the switch from day to night. We agree it is currently unclear what exactly is being timed.

2. Is there a typo in line 68, where it is stated that 390 μ M theophylline was used, whereas it is 370 in the figure?

Thanks, this was a typo.

3. In Figure 2C, the mean peak-to-peak time changes by about 30% in low copy number conditions. I found this quite interesting. Could the authors briefly comment on this effect?

We also found this interesting, but don't have a simple mechanistic explanation. In general, the mean period of an oscillator may increase or decrease in the stochastic limit. Notably, the mathematical model of the Kai system (in Fig. 3) shows a similar increase of the period as copy number increases. We've added the following note to the text:

We quantified the distribution of peak-to-peak times in these single cell movies (Fig. 2C), and found that, in addition to increased variability, the mean period of the oscillation increases at low expression levels. Both of these effects would tend to make the clock state mismatch the 24 hour day-night cycle.

4. On fitting timing error to KaiC levels while varying the number of phosphorylation steps (line 105 and Extended

data figure 7): presumably there can be up to 12 steps because there are two phosphosites in each KaiC monomer. Can the authors make this clearer for those not familiar with the system, as at first it is slightly confusing that Figure 3 shows what appear to be a maximum of 6 steps? Also, while the y-axis in the supplementary figure is labeled as coefficient of variation, the same quantity is called timing error elsewhere. Perhaps it would be better to maintain consistency.

We analyzed several mathematical models of the KaiABC oscillator and found that the stochastic scaling of these models differ (see our response to Reviewer 3 below). The generalized model we present in this paper allows us to vary the number of reaction steps in each hexamer that are required to toggle between a KaiB-binding state and a KaiB-nonbinding state, which is an important determinant of stochastic effects. Thus, the model does not specify the total number of phosphorylations on a hexamer in an explicit sense—only those that contribute towards this KaiB-binding toggle switch. The reviewer is correct that 12 is the maximum based on the biochemical features of the system, and would correspond to the requirement that KaiC cycles between completely phosphorylated and completely unphosphorylated states in order to bind and unbind KaiB. Empirically, the amplitude is smaller than this—the peak is not fully phosphorylated, and some phosphorylation remains at the trough. In any case, we thank the reviewer for bringing this point to our attention, and we have clarified this point in the main text:

Because the model we implement here allows us to vary the number of reaction steps between the peak and trough while holding other properties constant, we explored stochasticity as a function of step number, with five steps giving the best match to the experimental data

Thanks for the comment of “timing error”, we’ve modified the axis label in the supplementary figure to be consistent.

5. Can the authors expand a bit more on the necessity of including the C* state in their model (extended data figure 5)? Couldn't the ground state itself undergo non-KaiA mediated phosphorylation at an appropriately lower rate?

The difference between the states is that one is bound to KaiA, and the C state phosphorylates independently of KaiA. The reason these have to be separately accounting for is that the KaiA•KaiC complexes lower the pool of free KaiA.*

6. In Supplementary methods, under “Determining model timing error”, the ~24h doubling time is referenced as “data not shown”, which is to be avoided as per Nature Comm guidelines. No further figure is needed in my opinion, the authors could just delete that passage.

Thanks, we’ve deleted this line.

7. Line 219 – Spelling mistake - YPF rather than YFP.

Thanks!

Reviewer #3 (Remarks to the Author):

The manuscript, “Demand for high protein copy number can favor timers over clocks in bacteria,” by Rust et al. introduces a clever hypothesis for the evolution of oscillator vs. hourglass timer mechanisms for biological timekeeping in cyanobacteria. As I understand it, the authors argue that cyanobacteria start out with a kaiABC oscillator system, but as some species (esp. Prochlorococcus) pursue a streamlining strategy that includes smaller and smaller cell sizes, at some point the number of Kai proteins becomes too small to support phase coherent oscillations and downgrades to a timer system, at which point KaiA is no longer needed and is (mostly) deleted from the genome. This is an imaginative idea that provides a potential selective pressure for the loss of kaiA from the Prochlorococcus genome, in contrast to the other scenario that has been proposed (Axmann et al. J. Bacteriol. 2009; Johnson et al., Nature Reviews Microbiol. 2017), namely that KaiA was lost from the Prochlorococcus genome as part of genome streamlining so that a timer resulted, and that in the consistent & regular environment of the ocean, a timer was not a liability and therefore was not selected against.

Having said that, there are a number of defects with the current manuscript that lessen the attractiveness of the study. To warrant publication in Nature Communications, the authors need to perform new experiments and undertake a major revision of the current manuscript.

The first part of the paper pursues a strategy in *S. elongatus* of reducing kai protein levels and finding a threshold for kai protein levels that oscillates and comparing this with the levels measured in the cells. Three items regarding this part of the paper:

1. If the data of Figures 1-3 were the only experimental support for the overall hypothesis, it would not suffice to make

the point of stochastically limiting number of molecules. The data of Figures 1-3 can be interpreted merely as a concentration effect, and we know well that the in vitro oscillation will also fail at low protein CONCENTRATION when the NUMBER of molecules is not limiting. Therefore, to make their point of limiting NUMBER of molecules, the Prochlorococcus case is necessary to be persuasive that the effects observed by the authors are not merely due to concentration. See below for concerns about the Prochlorococcus data.

We thank the reviewer for bringing up this point. It is of course correct that lowering protein expression while keeping the cell volume the same decreases both protein copy number and protein concentration. To disentangle these effects, we compared cells of different sizes at the same inducer level (shown in Figure 3D). In general, smaller cells have decreased coherence at the same inducer levels, suggesting that absolute copy numbers play a role.

We have also now added an analysis of the concentration dependence of the purified KaiABC oscillator (Extended Data Fig 4) where we monitor oscillations using a fluorescent KaiB probe. Indeed, oscillations fail at sufficiently low concentrations, but robust ~24 hour oscillations persist down to 1.2 μ M KaiC vs. our estimate of 2.1 μ M KaiC in vivo at our low theophylline conditions. This further suggests that the irregular oscillations in cells are not due to concentration effects.

2. The methodology for independently controlling KaiA and KaiBC levels is imaginative and will certainly be useful for other studies in a variety of labs, so the authors are to be congratulated for designing this expression methodology.

Cheers!

3. The measurements for KaiA concentration differ from those published by Kitayama et al. in 2003 (KaiB and KaiC concentrations are similar). More information is needed here as this is a crucial point for their interpretations and their measurements are different from those of Kitayama and co-authors. In particular, please provide on Extended Data Figure 1 the following: panels A and C should label the amount of the purified Kai protein loaded on each lane so we don't have to guess at the intermediate amounts, and show the plots of the standard curves with error bars for the replicate measurements. Moreover, the authors do not refer to the Kitayama et al. paper—why not? It is fundamental scientific courtesy to refer to the prior measurements, and surely the authors know about that important publication. Finally, the authors should address why their results differ from the results of Kitayama and coworkers for WT cells.

Thanks for this comment. It was an accidental oversight on our part to omit the Kitayama 2003 citation from the initial submission. Reviewer 1 raised this point as well, and we have modified the text to include more discussion of agreement and discrepancies between our measurements and the Kitayama study.

We have made the suggested changes to the labels in Extended Data Figure 1 and included plots of our standard curves.

Moreover, measurements of protein abundance need to be performed under the same metabolic conditions as the rhythm analyses, i.e., agar not liquid cultures. Metabolic state varies dramatically in cyanobacteria under agar vs. liquid cultures.

We took steps to match overall protein expression levels between liquid culture and agar pad microscopy conditions, by adjusting light levels to match single cell EYFP fluorescence intensity, a process we describe in the supplementary methods under "Identifying functionally equivalent light levels across experimental setups". Unfortunately, it is not feasible to obtain enough cellular material from the exact culture conditions used on the microscope to use for Western blotting.

Regarding the data obtained with Prochlorococcus cells, the authors MUST specify which strain of Prochlorococcus marinus they are studying. There is a large range of sizes among different Prochlorococcus strains (Cermak et al. ISME J. 11: 825-828, 2007), and in some cases there is only a small difference between the volumes of Prochlorococcus cells and Synechococcus cells (e.g., Bertilsson et al. Limnol. Oceanogr., 48: 1721–1731, 2003). The authors should show actual micrographs of the cells they are comparing as a helpful addition to the diagrams in Figure 4A.

We are using P. marinus MED4. We have now included electron micrographs of both the S. elongatus and P. marinus cells used in our experiments (Extended Data Figure 7), which are consistent with previously reported estimates of cell volume. We've modified the following text to clarify this:

we focused on the small cyanobacterium *Prochlorococcus marinus MED4*, whose cell volume is over twenty times smaller than *S. elongatus* (Extended Data Fig. 7)

Regarding the model, three concerns arise:

1. Why does the KaiC dephosphorylate in the dark in *Prochlorococcus*? Is it because of declining ATP:ADP in the dark? If so, the authors need to measure [ATP], [ADP], and [AMP] in the light & dark in *Prochlorococcus*, and express it as energy charge, as done by Puszynska & O'Shea (Elife 6: e23210, 2017). Also in that context, Puszynska & O'Shea did NOT find a significant change in energy charge in darkness in *Synechococcus*, in apparent contradiction to the previous ATP:ADP results of the current authors. How do the authors reconcile the results of Puszynska & O'Shea with their own previous results and with the current model?

Using the ATP/ADP ratio as a proxy for the input signal in the Prochlorococcus model is purely a hypothesis. The conclusions about the ability of the clock vs timer versions of the Kai system to tolerate noise don't depend on the biochemical mechanism of input, only that it affects KaiC phosphorylation. We have added a note to the main text to clarify this:

Though the mechanism of environmental input to the *P. marinus* Kai system is not known, we model the input as a modulation of KaiC phosphorylation rates, similar to the effect the ATP/ADP ratio has on KaiA-dependent KaiC phosphorylation in the *S. elongatus* system.

There is no evidence to support any molecular mechanism for dephosphorylation of Prochlorococcus KaiC in the dark (indeed, our study is to our knowledge the first report of KaiC phosphorylation in Prochlorococcus). The hypothesis that the ATP/ADP ratio is involved is reasonable, but untested. It will be an interesting future direction to understand the biochemistry of the Prochlorococcus Kai proteins in detail.

*It is beyond the scope of this study, but we note that in our hands in *S. elongatus*, the magnitude of both changes in energy charge in the dark and the extent of oscillator phase shifts depend on culture conditions (but are correlated), (for example illumination intensity, see Pattanayak et. al 2014, Fig. S4).*

2. The authors use a "simplified" mathematical model in the presentation of Figure 4. However, the senior author is well known for his modeling of the clock mechanism of cyanobacteria on the basis of both phosphorylation sites, etc. Why did the authors revert to a "simplified" model for this paper, when the authors could have easily use the more complicated model and refer to their previous papers? Does the more accurate and complicated model NOT show the effects depicted in Figure 4? I would like to see the essential simulation results repeated with the more accurate model and shown in the Supplement.

We have extended our analysis of KaiABC post-translational oscillator models to include stochastic versions of the models from Rust et al. 2007, Teng et al. 2013, and Lin et al. 2014. A general observation here is that, though all of these models can generate circadian oscillations in the deterministic limit, their scaling with protein copy number in the stochastic regime are quite different. The Rust et al. model predicts oscillations that are less irregular than our microscopy observations; the Lin et al. model predicts oscillations that are more irregular. In this sense, the reviewer is correct that previous models make inconsistent stochastic predictions.

There are at least two important determinants of the strength of stochastic effects. One is whether the model describes monomeric subunits acting independently (as in Rust et al.) or hexamers acting cooperatively (more realistic, as in Lin et al.). Another is the number of irreversible reaction steps in each hexamer between the peak and trough of the oscillation. We deliberately devised the model in this manuscript to allow this property (reaction steps per cycle) to be tuned to attempt to get agreement with data. Going forward, demanding the correct stochastic performance from models will likely be an important criterion for distinguishing models.

We've added a note to the main text to try to clarify our approach to mathematical modeling of the stochastic system:

Though many models of the Kai oscillator can produce circadian rhythms when the role of molecular noise is ignored, we find that different models give striking different estimates of the magnitude of stochastic effects (Extended Data Fig. 6C). We suspected that, in addition to the degree of cooperativity in a model, the impact of stochasticity depends on the number of steps in the phosphorylation cycle required to switch between phosphorylation and dephosphorylation. Because the model we implement here allows us to vary the number of reaction steps between the peak and trough while holding other properties constant, we explored stochasticity as a function of step number, with five steps giving the best match to the experimental data (Extended Data Fig. 7, Fig. 3D)

3. In the context of issue # 2 above, the authors' hypothesis crucially depends upon the enhancement of phase

coherence with a timer at low copy number, as shown in Figure 4E. And indeed, the timer does better at low copy number (“450” NkaiC). However, the timer also appears to have better phase coherence in constant light at HIGH copy number (“14,400” NkaiC). This should not be the case if the negative feedback system is operating appropriately, and this comparison of the constant light simulations suggest that the model has been over-tweaked to obtain the desired result at low copy number.

Indeed, the reviewer is correct that the timer system (without a feedback loop) shows less vulnerability to molecular noise even at high copy number compared to the free-running oscillator. This is not a consequence of fine-tuning or over-tweaking of parameters but is a general phenomenon. In this paradigm, where performance of Kai system is defined as its ability to correctly predict the time in 12:12 LD cycles, the free-running oscillator only shows an advantage when there are random fluctuations in the environment. We have added additional analysis in Extended Data Figure 10 to clarify. We agree that the initial presentation of Fig 4E and 4F was confusing with regards to whether environmental fluctuations were present or absent, and we have clarified this in the text (see above response to Reviewer 2).

Prochlorococcus KaiC phosphorylation oscillates in light/dark but not in constant conditions, thereby exhibiting what appears to be an hourglass timer. It is the hypothesis of this manuscript that an hourglass timer was selected because the number of Kai molecules became too small. But the alternative hypothesis suggested by other labs is that KaiA was lost, creating a timer and it didn't matter to the Prochlorococcus in the constant marine environment. In this context:

1. knowing if a kaiA-deletion strain of *S. elongatus* exhibits a KaiC phosphorylation rhythm in light/dark but not in constant light could enhance the authors' arguments.

*We are not aware of measurements of KaiC phosphorylation in a kaiA-null strain in LD cycles, but we think it is unlikely because of the seemingly absolute requirement for KaiA to be present in order to observe sustained phosphorylation in vivo or in vitro. Notably, *P. marinus* KaiC has diverged from the other cyanobacterial KaiC in the C-terminal tail (putative KaiA interaction site) and the A-loop which mediates KaiA-dependent control of kinase activity, first pointed out by Holtzendorff et al. Thus, it is likely that *P. marinus* KaiC has fundamentally changed how its kinase activity is regulated vs. *S. elongatus*.*

2. the authors should acknowledge the previous discussions of the alternative model (papers mentioned above in the first paragraph of this review), and discuss the pros and cons of each evolutionary scenario.

*Thanks, we appreciate the perspective here. Certainly, we do not have evidence for the evolutionary drivers that led to changes in the Prochlorococcus kai system, and we cannot conclusively state that the copy number constraints were the dominant reason for the deletion of kaiA. Given that the system exists at low protein copy number in *P. marinus*, our conclusion is that there may in fact be functional advantages to losing the free-running property of the KaiABC oscillator beyond genome streamlining. We have added a few sentences to the main text as suggested:*

****P. marinus* apparently diverged from a common ancestor with marine *Synechococcus* that has a *kai* gene cluster with all three genes intact. It has undergone both a reduction of cellular volume and a general genome streamlining that includes the loss of *kaiA*. In light of our analysis, the loss of *kaiA* and the loss of stable rhythmicity may not only reflect the benefits of genome reduction, but may also be an adaptation that outperforms a free-running clock in a low protein copy number niche.***

A few minor points that should be simple to correct:

1. Lines 81-83: the authors should clarify that timing error of small cells increases at LOW induction levels. Their point is confusing as currently written.

Thanks, we've rewritten this as:

“We quantified relative peak-to-peak timing errors in these cells at different induction levels and found that shorter cells had noisier rhythms compared to longer cells at low induction conditions (Fig. 2D).”

2. Fig. 2A and accompanying movies: from the movies, it appears that the loss of phase coherence appears to occur early in the growth of the micro-colonies and appears to stabilize thereafter. From the authors' analyses, is this observation true? And if so, how do the authors explain this factor in their overall interpretations?

Thanks for raising this point. Because the distributions of peak-to-peak timing are calculated across all cell lineages, in a growing population, most data points come from the latter part of the micro-colony growth. Because of this, we don't have sufficient statistical power to decide whether there is a difference in stochasticity between the beginning and end of the movie, but certainly stochastic effects are still present during the last couple of days.

3. Fig. 3B: how is the %phosphorylation calculated? Why is it so low? These are not physiological values, and may indicate a problem with the model.

4. Fig. 4E: same concerns about %phosphorylation as in #3 above (for Fig. 3B).

Thanks for bringing this to our attention. Because this model does not explicitly describe the phosphorylated residues and is instead meant to capture the number of reaction steps per hexamer between the peak and trough, it was inappropriate for us to label these plots "% phosphorylation". What's actually plotted is the average position of the population of molecules in the reaction loop, which can't be easily converted to a quantity that can be directly experimentally measured. We've relabeled these plots "KaiC state" and defined this in the figure caption.

5. A parameter set for the numerical simulations is not given in the supplement.

We've now added this.

6. On lines 120 & 121, the authors claim that reference # 18 supports their postulate that the KaiABC complexes represent only a small fraction of total KaiC, but in fact the data in that paper does NOT support this interpretation (Figure 2 of Kageyama et al. 2006 may appear to support the claim, but Figure 3 of that paper clearly indicates that the KaiABC complex is a large proportion of the total KaiC pool at incubation time 36).

We agree that the Kageyama 2006 data do not directly support this interpretation. It's apparent from the Kageyama paper that a large fraction of KaiC is in a high molecular weight fraction that also contains KaiB and KaiA, but how many KaiC molecules are bound to KaiA and KaiB (vs. just KaiB) is not clear. This claim in the text is fundamentally about how stochasticity arises in the model and future direct experimental tests would be illuminating. We've removed the citation from this sentence.

7. In Extended Data Figure 9, it is difficult to see how the authors could have derived the clear oscillations of %P-KaiC densitometry plots from the fuzzy immunoblots in which the bands of phospho- vs. non-phospho-KaiC are not resolved well.

The reviewer is correct that bands cannot be clearly resolved in these blots. We analyzed the blots by manually drawing a cutoff (based on the gel shift pattern of purified protein) across the blot and taking the ratio of signal above and below the line. The Extended Data Figure now shows this line. Because there is a degree of arbitrariness here, we've relabeled the axes of the quantification plots "estimated %P-KaiC".

REVIEWERS' COMMENTS:

Reviewer #1 (Remarks to the Author):

I'm very pleased to see that the authors addressed my major and minor concerns comprehensively. Thus, the manuscript has improved greatly and would be suitable for publishing at Nature Communications. However, one of my major concerns remained: The protein-quantification procedure is still lacking detail which is needed for reproducibility. I'd recommend to provide in a Suppl. file the raw data - intensities derived from protein staining - as well as the formulas used for calculating the protein weights [ng] and the corresponding KaiA, KaiB and KaiC molecule numbers.

Reviewer #2 (Remarks to the Author):

The authors have addressed all my concerns fully. They have done an excellent job of responding to all the reviewer comments, and I support publication in Nature Communications.

Reviewer #3 (Remarks to the Author):

The revised manuscript, "Demand for high protein copy number can favor timers over clocks in bacteria," by Rust et al. is much improved. While it does not totally satisfy the concerns that I expressed regarding the original submission, I agree the authors have diligently and conscientiously attempted to address the issues raised.

There is one deficiency that the authors did not address adequately, namely the presentation of the alternative hypothesis proposed by Axmann et al. (J. Bacteriol. 2009) that "the observed evolutionary reduction of the clock locus in *Prochlorococcus* are consistent with a model in which a mechanism that is less robust than the well-characterized KaiABC protein clock of *Synechococcus* is sufficient for biological timing in the very stable environment that *Prochlorococcus* inhabits." As requested in the review of the original submission, this alternative hypothesis should be presented clearly (and Axmann et al. referenced in the bibliography) in the area of lines 210-215 and should not be skirted as the authors have done in the revised manuscript.

Moreover, there are a few remaining issues from the first review that I ask the authors to consider for their final revisions and/or future studies:

1. Cyanobacteria growing in liquid culture vs. on agar are in two very different metabolic states, and it is not really appropriate to switch between these states for measurements and monitoring. It almost smacks of picking and choosing data from two different sources to support the researchers' models/hypotheses.
2. Thanks for showing the EM photos in Extended Data Figure 7, but those photos suggest a volume difference of 5-10X, certainly not a 20X difference as the authors contend on line 175.
3. The statement, "However, our analysis suggests that there is a minimum biosynthetic investment needed to create a reliable oscillator. Microbial cells span a wide range of sizes, and for very small cells, expressing many thousands of copies of clock proteins may not be tenable. This suggests that tiny cells may use alternative dynamical strategies to keep time." may be an overstatement. For example, the eukaryote *Ostreococcus tauri* can maintain a circadian clock even with its 0.8 μ m size. Since it is only slightly bigger than the authors' strain of *Prochlorococcus* (and the cytoplasmic or nuclear compartments may independently be smaller than the volume of a

Prochlorococcus cell), it may be an overstatement to say that small size necessitates alternative timing mechanisms.

4. In their Response, the authors statement "our study is to our knowledge the first report of KaiC phosphorylation in Prochlorococcus" is incorrect. Such data were reported in Nature Reviews Microbiology 15: 232-242 (2017), Figure 4. However, that is a review paper and the raw data were not shown, so it is not necessary for the authors to acknowledge it in the current manuscript. Nevertheless, they should know about it.

5. The authors' comment regarding modeling in the Response about the "determinants of the strength of stochastic effects" may be problematic for interpreting the modeling reported in previous publications from this group. Nevertheless, their comparison in Extended Data Figure 6 is very helpful.

Reviewer #3 (Remarks to the Author):

The revised manuscript, "Demand for high protein copy number can favor timers over clocks in bacteria," by Rust et al. is much improved. While it does not totally satisfy the concerns that I expressed regarding the original submission, I agree the authors have diligently and conscientiously attempted to address the issues raised.

There is one deficiency that the authors did not address adequately, namely the presentation of the alternative hypothesis proposed by Axmann et al. (J. Bacteriol. 2009) that "the observed evolutionary reduction of the clock locus in *Prochlorococcus* are consistent with a model in which a mechanism that is less robust than the well-characterized KaiABC protein clock of *Synechococcus* is sufficient for biological timing in the very stable environment that *Prochlorococcus* inhabits." As requested in the review of the original submission, this alternative hypothesis should be presented clearly (and Axmann et al. referenced in the bibliography) in the area of lines 210-215 and should not be skirted as the authors have done in the revised manuscript.

The hypothesis that a very regular environment permits non-free-running timing systems is not mutually exclusive with our findings about the role of internal noise. Indeed, our mutual information calculations suggest that both play a role. We've added the following to the Discussion section to clarify:

*"Because *Prochlorococcus* species are typically found in near-equatorial waters of the open ocean, previous studies have suggested that the hypothesis that a free-running clock may be dispensable because of the high regularity of environmental cycles. This hypothesis about the external environment is consistent with our conclusion that high levels of internal noise disfavor free-running clocks. Indeed, in our numerical simulations, free-running behavior is most favored when both internal noise is low and when the external environment has large irregular fluctuations (Supplementary Fig. 10). Thus, both of these effects may have contributed to the loss of free-running rhythms in *P. marinus*."*

Moreover, there are a few remaining issues from the first review that I ask the authors to consider for their **final revisions** and/or future studies:

1. Cyanobacteria growing in liquid culture vs. on agar are in two very different metabolic states, and it is not really appropriate to switch between these states for measurements and monitoring. It almost smacks of picking and choosing data from two different sources to support the researchers' models/hypotheses.

We understand the Reviewer's point that different growth conditions lead to different physiological states. We did as much as we were able to match conditions between agar and liquid. The simple reason is that we have no way to monitor single cells in liquid, and, conversely, no way to collect enough material for immunoblotting from agar pad cultures.

2. Thanks for showing the EM photos in Extended Data Figure 7, but those photos suggest a volume difference of 5-10X, certainly not a 20X difference as the authors contend on line 175.

Quantifying the cytoplasmic volume from a two-dimensional projection is not completely straightforward. In any event, our measurements of protein copy number per cell do not depend on cell volume because they are calculated as ratios of protein in lysate to total number of cells lysed.

3. The statement, "However, our analysis suggests that there is a minimum biosynthetic investment needed to create a reliable oscillator. Microbial cells span a wide range of sizes, and for very small cells, expressing many thousands of copies of clock proteins may not be tenable. This suggests that tiny cells may use alternative dynamical strategies to keep time." may be an overstatement. For example, the eukaryote *Ostreococcus tauri* can maintain a circadian clock even with its 0.8µm size. Since it is only slightly bigger than the authors' strain of *Prochlorococcus* (and the cytoplasmic or nuclear compartments may independently be smaller than the volume of a *Prochlorococcus* cell), it may be an overstatement to say that small size necessitates alternative timing mechanisms.

*We agree that the copy number scaling we observe may be particular to the KaiABC system. It is possible that *Ostreococcus* uses an oscillator that performs better at low copy number, or that it can afford to highly express key clock proteins to achieve high copy numbers. These are interesting directions for future work.*

4. In their Response, the authors statement "our study is to our knowledge the first report of KaiC phosphorylation in *Prochlorococcus*" is incorrect. Such data were reported in *Nature Reviews Microbiology* 15: 232-242 (2017), Figure 4. However, that is a review paper and the raw data were not shown, so it is not necessary for the authors to acknowledge it in the current manuscript. Nevertheless, they should know about it.

Thanks for reminding us of this review figure. We've added a citation to it.

5. The authors' comment regarding modeling in the Response about the "determinants of the strength of stochastic effects" may be problematic for interpreting the modeling reported in previous publications from this group. Nevertheless, their comparison in Extended Data Figure 6 is very helpful.

We agree that it important to consider the description of stochastic effects in clock models. Our previous work (and the work of other groups) has not paid sufficient attention to this. We've added this sentence to the Results section:

"...we find that different models^{13,17,20} give strikingly different estimates of the magnitude of stochastic effects, implying that stochastic behavior can be used to discriminate between alternative mechanisms (Supplementary Fig. 6C)."